# Stable-Pose: Leveraging Transformers for Pose-Guided Text-to-Image Generation

Jiajun Wang[*1], Morteza Ghahremani[*1,3], Yitong Li[*1,3], Björn Ommer[2,3], and Christian Wachinger[1,3]

[1]Lab for AI in Medical Imaging, Technical University of Munich (TUM), Germany
[2]CompVis @ LMU Munich, Germany
[3]Munich Center for Machine Learning (MCML), Germany

## Abstract

Controllable text-to-image (T2I) diffusion models have shown impressive performance in generating high-quality visual content through the incorporation of various conditions. Current methods, however, exhibit limited performance when guided by skeleton human poses, especially in complex pose conditions such as side or rear perspectives of human figures. To address this issue, we present Stable-Pose, a novel adapter model that introduces a coarse-to-fine attention masking strategy into a vision Transformer (ViT) to gain accurate pose guidance for T2I models. Stable-Pose is designed to adeptly handle pose conditions within pre-trained Stable Diffusion, providing a refined and efficient way of aligning pose representation during image synthesis. We leverage the query-key self-attention mechanism of ViTs to explore the interconnections among different anatomical parts in human pose skeletons. Masked pose images are used to smoothly refine the attention maps based on target pose-related features in a hierarchical manner, transitioning from coarse to fine levels. Additionally, our loss function is formulated to allocate increased emphasis to the pose region, thereby augmenting the model's precision in capturing intricate pose details. We assessed the performance of Stable-Pose across five public datasets under a wide range of indoor and outdoor human pose scenarios. Stable-Pose achieved an AP score of 57.1 in the LAION-Human dataset, marking around 13% improvement over the established technique ControlNet. The project link and code are available at `https://github.com/ai-med/StablePose`.

## 1 Introduction

Pose-guided text-to-image (T2I) generation holds immense potential for swiftly producing photo-realistic images that exhibit contextual relevance and accurate posing through the integration of text prompts and pose instructions. The kinematic or skeleton pose provides a set of key points (joints) that represent the skeletal framework of the human body (shown in Figure A.1). Despite the sparsity, skeleton-pose data offers sufficient details of human poses with high flexibility and computational efficiency for T2I generation in various applications such as animation, robotics, sports training, and e-commerce, making it user-friendly and ideal for real-time applications [21]. Juxtaposed with other forms of pose information like volumetric pose with dense content, skeleton pose is capable of conveying heightened articulation information, facilitating intuitive interpretation and flexible manipulation of human poses [28; 20].

Traditional pose-guided human image generation methods require a source image during training for dictating the style of the generated images [22; 23; 42; 38; 47]. Such methods, while offering

---

[*]Equal Contribution. `{jiajun.wang, morteza.ghahremani, yi_tong.li}@tum.de`

38th Conference on Neural Information Processing Systems (NeurIPS 2024).

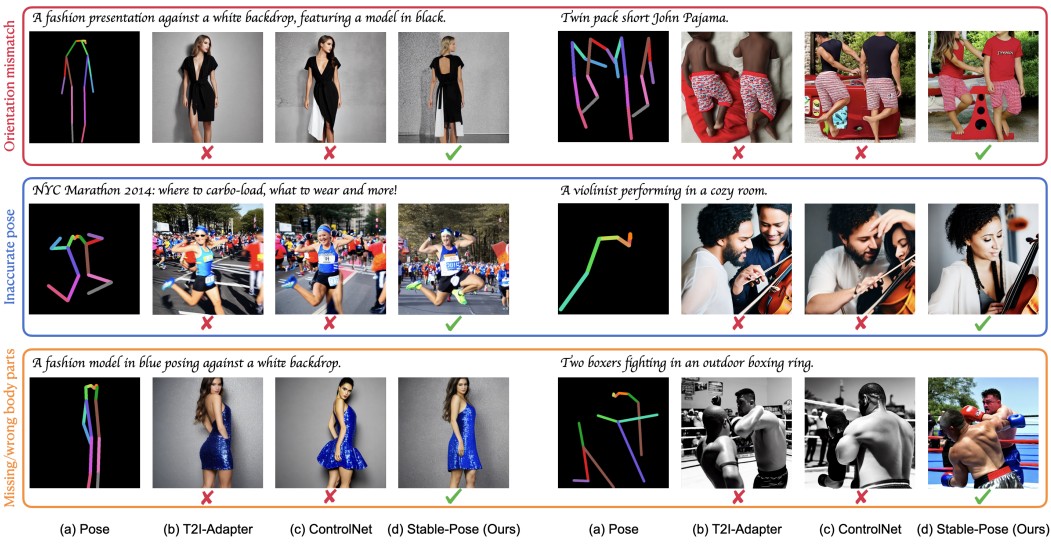

Figure 1: Stable-Pose leverages the patch-wise attention of ViTs to address the complex pose conditioning problem in T2I generation, showing superior performance compared to current techniques.

control over the appearance, limit the flexibility and diversity of the output and depend heavily on the need for paired source-target data during training. In contrast, recent advancements in controllable T2I diffusion models have shown the potential to eliminate the need for source images and allowed for higher creative freedom by relying on text prompts and external conditions [44; 46; 13; 25; 18]. Enabling more versatile visual content creation, these methods often face challenges in precisely aligning conditional images with sparse representations such as skeleton pose data, especially when dealing with complex pose scenarios like those depicting the back or side views of human figures (Figure 1). Moreover, existing methods may also fail to maintain accurate body proportions, resulting in an unnatural appearance of the body.

To address the insufficient binding of sparse pose data in T2I generation models, a potential strategy is to capture long-range patch-wise relationships among various anatomical parts of human poses. In this paper, we introduce *Stable-Pose* that integrates vision Transformers (ViT) into pre-trained T2I diffusion models like Stable Diffusion (SD) [33], with the goal of improving pose control by capturing patch-wise relationships from the specified pose. In Stable-Pose, the learnable attentions adhere to an innovative coarse-to-fine masking approach, ensuring that pose conditioning is directed toward the relevant pose areas while preserving the diversity of the overall image. To further enhance this effect, a pose-mask guided loss is introduced to optimize the fidelity of the generated images in adherence to the given pose instructions. We evaluated Stable-Pose across five distinct datasets, covering indoor and outdoor image/video datasets. Compared to the state-of-the-art methods, Stable-Pose achieved the highest accuracy and robustness in pose adherence and generation fidelity, making it a promising solution for enhancing pose control in T2I generation. We further performed comprehensive ablation studies to demonstrate the effectiveness of our design. In summary, our contributions are:

- Addressing the challenge of generating photo-realistic human images in pose-guided T2I by integrating a novel ViT, achieving highly accurate synthesis in pose adherence and image fidelity, even under challenging conditions.

- Introducing a hierarchical integration of pose masks for coarse-to-fine guidance, with a novel pose-masked self-attention mechanism and pose-mask guided loss function. Stable-Pose is designed as a lightweight adapter that can be easily integrated into any pre-trained T2I diffusion models to effectively enhance pose control.

- Stable-Pose effectively learns to preserve intricate human shape structures and accurate body proportions, achieving exceptional performance across five publicly available datasets, encompassing both image and video data.

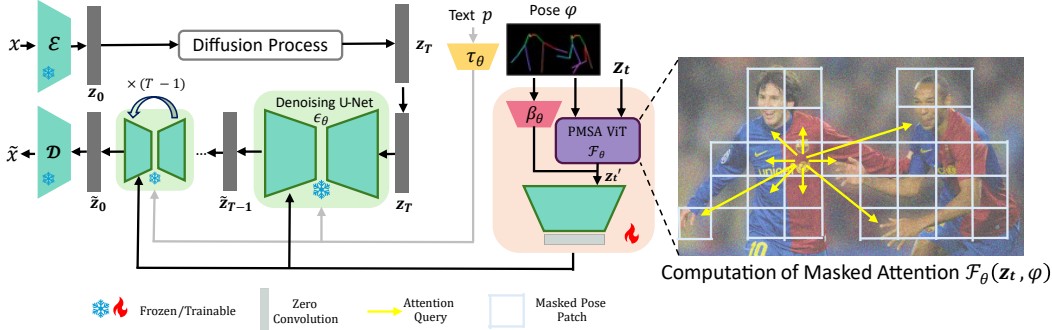

Figure 2: The Stable Diffusion architecture with Stable-Pose: operating on the pose skeleton image, Stable-Pose integrates a trainable ViT unit into the frozen-weight Stable Diffusion [33] to improve the generation of pose-guided human images.

## 2 Related Work

**Pose-Guided Human Image Generation**: Traditional pose-guided human image generation takes a source image and pose as input, aiming to generate photo-realistic human images in specific poses while preserving the appearance from the source image. Prior works [22; 23; 42; 38; 9] primarily utilized generative adversarial network (GAN) or variational autoencoder (VAE), treating the synthesis task as conditional image generation. Zhu et al. [47] integrated attention mechanism for appearance optimization in a Pose Attention Transfer Network (PATN). Zhang et al. [45] implemented a Dual-task Pose Transformer Network (DPTN), using a Transformer module to incorporate features from two tasks: an auxiliary source image reconstruction task and the main pose-guided target image generation task, thereby capturing the dual-task correlation. With the recent emergence of diffusion models [12], Bhunia et al. [4] proposed a texture diffusion module to transfer texture patterns from the source image to the denoising process. Moreover, classifier-free guidance [11] is applied to provide disentangled guidance for style and pose. Shen et al. [37] proposed a three-stage synthesis pipeline which progressively performs global feature extraction, target prediction, and refinement with three diffusion models. While the source image offers control over appearance, a limitation arises from the necessity of paired source-target data during training. Text, however, obviates this need and offers higher flexibility and diversity for the synthesis. Thus, incorporating text conditions for pose-skeleton-guided human image generation shows significant promise [19; 16].

**Controllable Diffusion Models**: Large-scale T2I diffusion models [33; 31; 32; 35; 26] excel at creating diverse and high-quality images, yet they often lack precise control with solely text-based prompts. Recent studies aim to enhance the control of T2I models using various conditions such as canny edge, sketch, and human pose [13; 44; 25; 18; 46; 16; 24; 29]. These approaches can be broadly classified into two groups: training the entire T2I model or developing plug-in adapters for pre-trained T2I models. As in the first group, Composer [13] trains a diffusion model from scratch with a decomposition-composition paradigm, enabling multi-control capability. HumanSD [16] fine-tunes the entire SD model using a heatmap-guided loss tailored for pose control. In contrast, T2I-Adapter [25] and GLIGEN [18] train lightweight adapters whose outputs are incorporated into the frozen SD. Similarly, ControlNet [44] employs a trainable copy of the SD encoder to encode conditions for the frozen SD. Uni-ControlNet [46] introduces a uni-adapter for multiple conditions injection to the trainable branch in a multi-scale manner. ControlNet++ [24] proposes to improve controllable generation by explicitly optimizing the cycle consistency between generated images and conditional controls. Our method aligns with the latter category, as it is characterized by reduced training time, cost-effectiveness, and generalizability.

## 3 Proposed Method

Stable-Pose has a trainable ViT unit that is integrated into the pre-trained T2I diffusion models to direct diffusion models toward the conditioned pose. In latent diffusion models (LDMs) [33], a pre-trained encoder $\mathcal{E}$ and decoder $\mathcal{D}$ are employed to transform an input RGB image $x \in \mathcal{R}^{H \times W \times 3}$ with height $H$ and width $W$ into a latent space with reduced spatial dimensions and vice versa. The diffusion process is then efficiently conducted in the down-scaled latent space. During the training, the forward process in diffusion models adds noise to the encoded RGB image $\mathbf{z}_0 = \mathcal{E}(x)$ to generate

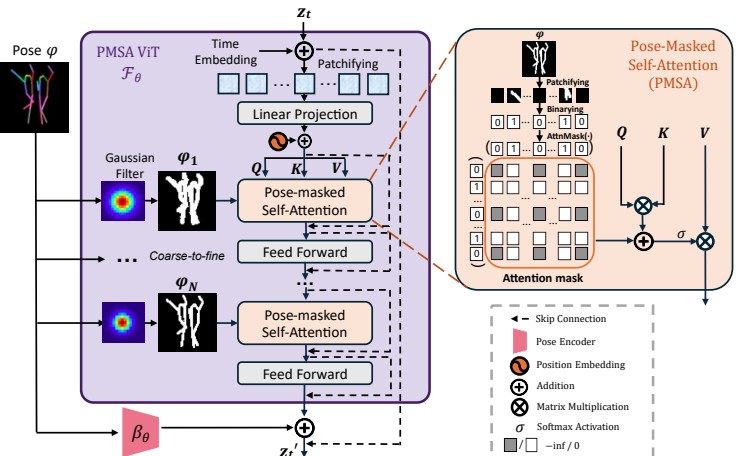

Figure 3: Stable-Pose consists of a pose encoder $\beta_\theta$ and a coarse-to-fine Pose-Masked Self-Attention (PMSA) ViT $\mathcal{F}_\theta$ for seeking the patch-wise relationship of human parts. PMSA restricts attention to embedding tokens within a specific attention mask to ensure that each embedding token can only attend to pose embedding tokens, not non-pose ones.

its noisy sample $\mathbf{z}_t \in \mathcal{R}^{C \times h \times w}$ with height $h$, width $w$, and channel $C$ via

$$\mathbf{z}_t = \sqrt{\bar{\alpha}_t}\mathbf{z}_0 + \sqrt{1-\bar{\alpha}_t}\epsilon, \quad \epsilon \sim \mathcal{N}(0, I), \quad t = 1, \cdots, T, \tag{1}$$

where $\bar{\alpha}_t$ is a pre-determined hyperparameter that controls the noise level at step $t$. The reverse process of diffusion models, so-called denoising, learns the statistics of the Gaussian distribution at each time step. The reverse process is formulated as:

$$p_\theta(\mathbf{z}_{t-1}|\mathbf{z}_t) = \mathcal{N}(\mathbf{z}_{t-1}; \mu_\theta(\mathbf{z}_t, t), \Sigma_\theta(\mathbf{z}_t, t)). \tag{2}$$

As shown in Figure 2, the denoising network $\epsilon_\theta$ adopts a UNet backbone, which is equipped with Stable-Pose for augmenting the latent encoding given the input conditional pose image. Let $\epsilon_\theta(\mathbf{z}_t, t, p)$, $t \in \{1, \cdots, T\}$, represent a T-step denoising UNet with gradients $\nabla\theta$ over a batch and input text prompt $p$. The conditional LDM is learned through $T$ steps by minimizing $\mathcal{L} = \mathbb{E}_{\mathbf{z},p,\varphi,\epsilon \sim \mathcal{N}(0,I),t} \left[ ||\epsilon - \epsilon_\theta(\mathbf{z}_t, t, \tau_\theta(p), \upsilon_\theta(\mathbf{z}_t, \varphi))||_2^2 \right]$, where Stable-Pose $\upsilon_\theta$ conditions the latent encoding $\mathbf{z}_t$ on the input pose skeleton $\varphi \in \mathcal{R}^{h \times w \times 3}$, and $\tau_\theta$ is a text encoder that maps the text prompt $p$ to an intermediate sequence. The proposed framework is detailed in Figure 3. Stable-Pose aims at improving the frozen decoder in the UNet of SD to condition the input latent encoding $\mathbf{z}_t$ on the conditional pose image $\varphi$:

$$\mathbf{z}_t' = \mathbf{z}_t + \upsilon_\theta(\mathbf{z}_t, \varphi). \tag{3}$$

In Stable-Pose, the pose image $\varphi$ and the given latent encoding $\mathbf{z}_t$ are processed by two main blocks named Pose-Masked Self-Attention (PMSA) $\mathcal{F}_\theta$ and pose encoder $\beta_\theta$ in such a way that

$$\upsilon_\theta(\mathbf{z}_t, \varphi) = \mathcal{F}_\theta(\mathbf{z}_t, \varphi) + \beta_\theta(\varphi), \tag{4}$$

where the pose encoder $\beta_\theta$ provides high-level features for the input pose while PMSA $\mathcal{F}_\theta$ explores the patch-wise relationship across input $\mathbf{z}_t$ using a self-attention mechanism and the binary-masked version of the pose image. PMSA employs a coarse-to-fine framework that provides additional guidance to the latent encoding, directing it towards attending to the conditioned pose. We detail each block in the subsequent sections. The updated latent encoding $\mathbf{z}_t'$ is subsequently fed through the encoder of SD, followed by a series of zero convolutional blocks, a structural resemblance to the architecture employed in ControlNet [44] for ensuring a robust encoding of conditional images.

**Pose encoder** $\beta_\theta : \mathcal{R}^{H \times W \times 3} \to \mathcal{R}^{f \times h \times w}$ is a trainable encoder that maps input pose skeleton image into a feature with height $h$, width $w$, and channel $f$. To this end, we employ a combination of six convolutional layers with SiLU activation layers [8], downsampling the input pose image by a factor of 8. A zero-convolutional layer is added in the end. As the input pose image contains sparse information, this straightforward pose encoder is sufficient for accurate encoding of the skeleton pose.

**PMSA** $\mathcal{F}_\theta : \mathcal{R}^{f_{in} \times h \times w} \to \mathcal{R}^{f \times h \times w}$ seeks the potential relationships between patches within the latent encoding $\mathbf{z}_t$. The interconnections of various parts of the human body suggest the presence of

---

**Algorithm 1** Generation of the attention mask for PMSA

---

1: **procedure** ATTNMASK($m_k$)
2:     $m_k$: Patchified and binarized pose masks of size $(N, H, W)$
3:     $m_k \leftarrow$ Flatten $m_k$ along the last two dimensions
4:     $N \leftarrow m_k.shape[0]$
5:     $L \leftarrow m_k.shape[1]$
6:     $attn\_mask \leftarrow$ initialize as tensor of size $(N, L, L)$ with entries set to $-inf$
7:     **for** $i \in [0, N-1]$ **do**
8:         $indices \leftarrow$ find indices where $m_k[i] == 1$
9:         $attn\_mask[i][indices, :] \leftarrow 0$
10:       $attn\_mask[i][:, indices] \leftarrow 0$
11:     **end for**
12:     **return** $attn\_mask$
13: **end procedure**

---

cohesive relationships among them. To capture this, we leverage the self-attention mechanism. We divide $\mathbf{z}_t$ into $L$ non-overlapping patches of size $p \times p$, i.e., $\mathbf{z}_t = \{\mathbf{z}_t^{(1)}, \ldots, \mathbf{z}_t^{(L)} | \mathbf{z}_t^{(l)} \in \mathbb{R}^{p^2 \times f_{in}}\}$. PMSA projects the patch embeddings $\mathbf{z}_t$ into Query $\mathbf{Q} = \mathbf{z}_t \mathbf{W}_Q$, Key $\mathbf{K} = \mathbf{z}_t \mathbf{W}_K$, and Value $\mathbf{V} = \mathbf{z}_t \mathbf{W}_V$ via three learnable weight matrices $\mathbf{W}_Q \in \mathbb{R}^{f_{in} \times f_q}$, $\mathbf{W}_K \in \mathbb{R}^{f_{in} \times f_k}$, and $\mathbf{W}_V \in \mathbb{R}^{f_{in} \times f_v}$, respectively. Then it computes the attention scores between all patches via

$$A_k = \texttt{attention}\left(\mathbf{Q}, \mathbf{K}, \mathbf{V}, \mathbf{m}_k\right) = \texttt{softmax}\left(\frac{\mathbf{Q}\mathbf{K}^\top}{\sqrt{f_q}} + \texttt{AttnMask}(\mathbf{m}_k)\right)\mathbf{V}. \tag{5}$$

In this equation, $\mathbf{m}_k$ denotes a binary mask derived from the input pose image, which is expanded by $\texttt{AttnMask}(\cdot)$ for being used in the self-attention computation (see Figure 3). $\mathbf{m}_k$ is obtained by converting the pose image into a binary mask and downsampling it into the same size of the latent vector $\mathbf{z}_t$. The resultant mask is then dilated by a Gaussian kernel of length $k$. The dilated-binary mask $\mathbf{m}_k$ is then partitioned into $L$ non-overlapping patches of size $p \times p$. The patches containing pose are labelled as 1 while others are marked as 0. We form a resulting $L \times L$ attention mask based on these $L$ patches through the function $\texttt{AttnMask}(\cdot)$. As illustrated in Algorithm 1, for patch entries that correspond to pose regions, both the respective row and column in the mask are set to 0. For all other regions not associated with pose, we assign an extremely small integer value. The attention mask helps to enhance the focus of PMSA on the pose-specific regions.

We implement a sequence of $N$ blocks of ViTs, each associated with a unique pose mask, arranged in a coarse-to-fine progression on the latent encoding. This approach gradually steers the latent encoding towards conforming to the specified pose condition. If $\boldsymbol{k} = \{k_1, \ldots, k_N\}$ denotes a set of Gaussian kernels where $k_1 > \cdots > k_N$, then the coarse-to-fine self-attention is obtained via

$$\begin{aligned} A_1 &= \texttt{attention}\left(\mathbf{Q}, \mathbf{K}, \mathbf{V}, \mathbf{m}_{k_1}\right) \\ A_n &= \texttt{attention}\left(\mathbf{Q}_{A_{n-1}}, \mathbf{K}_{A_{n-1}}, \mathbf{V}_{A_{n-1}}, \mathbf{m}_{k_n}\right), \quad \text{for } n = \{2, \cdots, N\}. \end{aligned} \tag{6}$$

Each encoding $A_n$ undergoes further processing by a Feed Forward unit, with the resulting $A_N$ integrated into the feature from the pose encoder $\beta_\theta$, as shown in Figure 3. The Feed Forward block consists of two linear transformations [3] combined with dropout layers, followed by ReLU non-linear activation functions [2].

**Pose-mask guided loss criterion**: The training of SD models requires high costs in hardware and datasets. Therefore, plug-in adapters for the frozen SD models, such as Stable-Pose, can enhance training efficiency by eliminating the need to compute gradients or maintain optimizer states for SD parameters. Instead, the optimization process focuses on improving Stable-Pose parameters. The loss in Stable-Pose aligns with the coarse-to-fine approach and is defined as follows:

$$\begin{aligned} \mathcal{L} = &\mathop{\mathbb{E}}_{\mathbf{z}, p, \varphi, \epsilon \sim \mathcal{N}(0, I), t}\left[|| \left(\epsilon - \epsilon_\theta\left(\mathbf{z}_t, t, \tau_\theta\left(p\right), \upsilon_\theta\left(\mathbf{z}_t, \varphi\right)\right) \odot \left(1 - \mathbf{m}_{k_N}\right)\right) ||_2^2\right] \\ &+ \alpha \mathop{\mathbb{E}}_{\mathbf{z}, p, \varphi, \epsilon \sim \mathcal{N}(0, I), t}\left[|| \left(\epsilon - \epsilon_\theta\left(\mathbf{z}_t, t, \tau_\theta\left(p\right), \upsilon_\theta\left(\mathbf{z}_t, \varphi\right)\right) \odot \mathbf{m}_{k_N}\right) ||_2^2\right]. \end{aligned} \tag{7}$$

Here, $\alpha \geq 1$ represents a predetermined pose-mask guidance hyperparameter that emphasizes the significance of masked region contents.

Table 1: Results on Human-Art dataset and LAION-Human subset. Methods with * are evaluated on released checkpoints.

| Dataset | Method | Pose Accuracy | | | Image Quality | | T2I Alignment |
|---|---|---|---|---|---|---|---|
| | | AP ↑ | CAP ↑ | PCE ↓ | FID ↓ | KID ↓ | CLIP-score ↑ |
| Human-Art | SD* | 0.24 | 55.71 | 2.30 | 11.53 | 3.36 | **33.33** |
| | T2I-Adapter | 27.22 | 65.65 | 1.75 | 11.92 | 2.73 | 33.27 |
| | ControlNet | 39.52 | 69.19 | 1.54 | 11.01 | **2.23** | 32.65 |
| | Uni-ControlNet | 41.94 | 69.32 | 1.48 | 14.63 | 2.30 | 32.51 |
| | GLIGEN | 18.24 | 69.15 | 1.46 | – | – | 32.52 |
| | HumanSD | 44.57 | 69.68 | **1.37** | **10.03** | 2.70 | 32.24 |
| | Stable-Pose (Ours) | **48.88** | **70.83** | 1.50 | 11.12 | 2.35 | 32.60 |
| LAION-Human | SD* | 0.73 | 44.47 | 2.45 | **4.53** | 4.80 | 32.32 |
| | T2I-Adapter* | 36.65 | 63.64 | 1.62 | 6.77 | 5.44 | 32.30 |
| | ControlNet* | 44.90 | 66.74 | 1.55 | 7.53 | 6.53 | 32.31 |
| | Uni-ControlNet | 50.83 | 66.16 | 1.41 | 6.82 | 4.52 | **32.39** |
| | GLIGEN | 19.65 | 66.29 | 1.40 | – | – | 32.04 |
| | HumanSD | 50.95 | 65.84 | **1.25** | 5.62 | 7.48 | 30.85 |
| | Stable-Pose (Ours) | **57.11** | **67.78** | 1.37 | 6.25 | **4.50** | 32.38 |

# 4 Experimental Results

**Datasets**. We assessed the performance of the proposed Stable-Pose as well as competing methods on five large-scale human-centric datasets including Human-Art [15], LAION-Human [16], UBC Fashion [43], Dance Track [40], and DAVIS [27] dataset. Details of the datasets and processing steps can be found in Sec. A.1.

**Implementation Details**. Similar to previous work [44; 25; 46], we fine-tuned our model on SD with version 1.5. We utilized Adam [17] optimizer with a learning rate of $1 \times 10^{-5}$. For our proposed PMSA ViT module, we adopted a depth of 2 and a patch size of 2, where coarse-to-fine pose masks were generated using two Gaussian filters, each with a sigma value of 3 but with differing kernel sizes of 23 and 13, respectively. We will explore the effects of these hyperparameters with more details in Sec. 4.2. In the pose-mask guided loss function, we set an $\alpha$ of 5 as the guidance factor. We also followed [44] to randomly replace text prompts as empty strings at a probability of 0.5, which aims to strengthen the control of the pose input. During inference, no text prompts were removed and a DDIM sampler [39] with time steps 50 was utilized to generate images. On the Human-Art dataset, we trained all techniques, including ours for 10 epochs to ensure a fair comparison. On the LAION-Human subset, we trained Stable-Pose, HumanSD [16], GLIGEN [18] and Uni-ControlNet [46] for 10 epochs, while we used released checkpoints from other techniques due to computational limitations. The training was executed using two NVIDIA A100 GPUs, with our method completing in 15 hours for the Human-Art dataset and 70 hours for the LAION-Human subset. This represents a substantial decrease in GPU hours compared to SOTA techniques. For instance, the T2I-Adapter requires approximately 300 GPU hours to train on a large-scale dataset. In contrast, our approach requires less than a quarter of that time and still delivers superior performance. We kept the same seed list for all techniques, including ours, during both training and inference time, to ensure fair comparison and reproducibility. More detailed information is provided in Sec. A.2.

**Evaluation Metrics**. We adopt six metrics for evaluation, covering pose accuracy, image quality, and text-image alignment. For pose accuracy, we employ mean Average Precision (AP), Pose Cosine Similarity-based AP (CAP) [1], and People Counting Error (PCE) [7], measuring the accuracy between the provided poses and the pose results extracted from the generated images by the pretrained pose estimator HigherHRNet [6]. For image quality, we use Fréchet inception distance (FID) [10] and Kernel Inception Distance (KID) [5]. Both metrics measure the diversity and fidelity of generated images and are widely used in image synthesis tasks. For text-image alignment, we include the CLIP score [30] that indicates how well the CLIP model believes the text describes the image. Details of the evaluation metrics can be found in Sec. A.4.

*Garage kits, a figurine of a girl with a gun on top of a small platform.*

*Digital art, a girl in a yellow dress is flying through the air with butterflies.*

*Bulldog boy and girl's track team's finish strong in Early Bird Meet.*

*Healthy yoga exercise on the beach, Stock Photo.*

| (a) Pose | (b) T2I-Adapter | (c) ControlNet | (d) Uni-ControlNet | (e) GLIGEN | (f) HumanSD | (g) Stable-Pose (Ours) |

Figure 4: Qualitative results of SOTA techniques and our Stable-Pose on Human-Art (first two rows) and LAION-Human (last two rows). An illustration of the pose input is shown in Figure A.1.

## 4.1 Results

**Quantitative and Qualitative Results**. Table 1 reports the quantitative results on both datasets among different methods. We reported the mean Average Precision (AP), Pose Cosine Similarity-based AP (CAP), People Count Error (PCE), Fréchet Inception Distance (FID), Kernel Inception Distance (KID), and the CLIP Similarity (CLIP-score). KID is multiplied by 100 for Human-Art and 1000 for LAION-Human for readability. Table 1 shows that Stable-Pose achieved the highest AP (48.87 on Human-Art and 57.41 on LAION-Human) and CAP (71.04 on Human-Art and 68.06 on LAION-Human), surpassing the SOTA methods by more than 10%. This highlights Stable-Pose's superiority in pose alignment. In terms of image quality and text-image alignment, Stable-Pose achieved comparable results against other methods, with only marginal discrepancy in FID/KID scores, yet the difference is negligible and the resulting quality remains high. Overall, these results underscore Stable-Pose's exceptional accuracy and robustness in both pose control and visual fidelity.

The qualitative results obtained from Human-Art [15] and LAION-Human [16] are illustrated in Figure 4. Consistent with the quantitative results, Stable-Pose demonstrates superior control compared to the other SOTA methods in both pose accuracy and text alignment, even in scenarios involving complex poses (the first row of Figure 4, which is a back view of the figure), and multiple individuals (the third row of Figure 4), while the other methods fail to consistently maintain the integrity of the original pose instructions. This is particularly evident in dynamic poses (e.g., yoga poses and athletic activities), where Stable-Pose manages to capture the pose dynamism more faithfully than others.

**Stable-Pose as a generic adapter**. Stable-Pose is designed as a lightweight generic adapter that can be easily integrated into any pre-trained T2I diffusion models to effectively enhance pose control. To further validate its generalizability, we conducted additional experiments by applying Stable-Pose on top of a pre-trained HumanSD [16] model. As shown in Table 2, the inclusion of Stable-Pose considerably improved the baseline HumanSD by over 10% in AP and 12% in KID, highlighting its high generalizability and effectiveness in enhancing both pose control and image quality.

**Results on Varying Pose Orientations**. Our experiments revealed that the current SOTA methods often faltered when tasked with creating images of humans in less common orientations, such as side or back poses. To investigate the capabilities of these methods in rendering atypical poses, we assembled a collection of around 2,650 images from the UBC Fashion dataset [43], which comprises exclusively front, side, and back poses. We evaluated the checkpoints of each technique from the LAION-Human dataset to assess pose alignment. As reported in Table 3, Stable-Pose significantly

Table 2: Stable-Pose as a lightweight adapter on pre-trained HumanSD model.

| Method | AP ↑ | CAP ↑ | PCE ↓ | FID ↓ | KID ↓ | CLIP-score ↑ |
|---|---|---|---|---|---|---|
| HumanSD | 44.57 | 69.68 | 1.37 | 10.03 | 2.70 | 32.24 |
| Stable-Pose | 48.88 | 70.83 | 1.50 | 11.12 | 2.35 | 32.60 |
| HumanSD+Stable-Pose | 49.24 | 71.01 | 1.42 | 10.42 | 2.37 | 32.16 |

outperforms other methods in recognizing and generating humans in all pose orientations, especially for rarer poses in side and back views, which surpasses the other methods by around 20% in AP. This further validates the robust controllability of Stable-Pose.

Table 3: Results of varying pose orientations on the UBC Fashion dataset. We report mean Average Precision (AP) across different methods for three orientations: front, side, and back.

| Orientation | T2I-Adapter | ControlNet | Uni-ControlNet | GLIGEN | HumanSD | Stable-Pose (Ours) |
|---|---|---|---|---|---|---|
| Front | 72.20 | 74.64 | 79.47 | 73.97 | 76.83 | $87.26_{9.80\%\uparrow}$ |
| Side | 36.80 | 52.83 | 58.26 | 45.32 | 57.09 | $69.76_{19.74\%\uparrow}$ |
| Back | 6.03 | 23.68 | 19.97 | 4.45 | 11.05 | $29.08_{22.80\%\uparrow}$ |

**Results on the Outdoor and Indoor Poses**. We extend the evaluation on both outdoor and indoor pose-guided T2I generation. We selected approximately 2,000 frames from the DAVIS dataset [27], which comprises videos of human outdoor activities, as our outdoor pose assessment. In addition, we randomly chose around 2,000 images from the Dance Track dataset [40], which is characterized by its group dance videos where most videos were shot indoors with multiple individuals and complex poses, as indoor pose-alignment evaluation. As shown in Table 4, the consistently highest AP and CAP scores achieved by Stable-Pose demonstrate its robustness in pose-controlled T2I generation across diverse environments, highlighting its potential as a backbone for pose-guided video generation. Further results can be found in Table A.6.

Table 4: Results on the outdoor poses from the DAVIS dataset and the indoor poses from the Dance Track dataset.

| Method | DAVIS | | Dance Track | |
|---|---|---|---|---|
| | AP ↑ | CAP ↑ | AP ↑ | CAP ↑ |
| T2I-Adapter | 20.28 | 60.76 | 10.36 | 72.38 |
| ControlNet | 30.13 | 60.81 | 16.45 | 73.16 |
| Uni-ControlNet | 37.64 | 60.57 | 25.22 | 73.62 |
| GLIGEN | 12.17 | 59.87 | 5.61 | 72.57 |
| HumanSD | 38.32 | 60.59 | 24.13 | 69.93 |
| Stable-Pose (Ours) | **42.87** | **62.43** | **28.58** | **74.97** |

## 4.2 Ablation Study

We conducted a comprehensive ablation study of Stable-Pose on the Human-Art dataset, including the effectiveness of pose masks, coarse-to-fine design, pose-mask guidance strength, and the effectiveness of PMSA and its ViT backbone. Further ablations on model parameters can be found in Sec. A.6.

**Effectiveness of Pose Masks**. To evaluate the impact of pose masks as input to our proposed PMSA and pose-mask guided loss function, we compared with removing them from the PMSA and/or from the associated loss function. As shown in Table 5, incorporating pose masks in both PMSA and loss function significantly enhanced the performance in both pose alignment and image quality.

Table 5: Results of the pose mask ablation on Human-Art dataset.

| Pose mask in PMSA | in loss | AP ↑ | CAP ↑ | PCE ↓ | FID ↓ | KID ↓ | CLIP-score ↑ |
|---|---|---|---|---|---|---|---|
| ✗ | ✗ | 39.40 | 69.18 | 1.55 | 13.94 | 2.56 | 32.63 |
| ✓ | ✗ | 44.50 | 70.51 | 1.51 | 14.24 | 2.61 | 32.58 |
| ✗ | ✓ | 45.39 | 70.18 | 1.56 | 13.17 | 2.62 | **32.65** |
| ✓ | ✓ | **48.88** | **70.83** | **1.50** | **11.12** | **2.35** | 32.60 |

**Coarse-to-Fine Masking Guidance**. The granularity of pose-masks is specified by the Gaussian kernels in Gaussian Filters, where a larger kernel generates coarser pose-masks. We compared the

results of constant granularity, fine-to-coarse as well as coarse-to-fine setting. All experiments are based on a ViT with PMSA and depth of 2 and Gaussian Filters with fixed sigma $\sigma = 3$. Details can be found in Sec. A.3. As indicated in Table 6, the coarse-to-fine approach consistently offers the best performance across metrics for pose alignment and image quality. This improvement is likely due to its progressive refinement from coarser to finer granularity in pose regions. By methodically narrowing the focus to more precise controllable areas, this strategy smoothly enhances the accuracy of pose adjustments and the overall quality of the generated images.

Table 6: Results of different Gaussian kernels $k$ used for pose masks in Stable-Pose.

| Pose mask granularity | AP ↑ | CAP ↑ | PCE ↓ | FID ↓ | KID ↓ | CLIP-score ↑ |
|---|---|---|---|---|---|---|
| Constant (23, 23) | 48.10 | 70.77 | 1.59 | 12.55 | 2.59 | **32.62** |
| Fine-to-coarse (13, 23) | 47.86 | **70.84** | 1.57 | 12.48 | 2.54 | 32.54 |
| Coarse-to-fine (23, 13) | **48.88** | 70.83 | **1.50** | **11.12** | **2.35** | 32.60 |

**Effectiveness of PMSA and its ViT backbone**. Our PMSA incorporates additional pose masks, derived from pose skeletons that have been expanded using Gaussian filters. To evaluate the effectiveness of PMSA, we instead only integrated these augmented pose masks into ControlNet without our PMSA block. We explored two configurations: one in which the original pose skeleton was concatenated with one coarsely enlarged pose mask, denoted as ControlNet-PM1, and another where it was concatenated with both the coarsely and finely enlarged pose masks, referred to as ControlNet-PM2. Table 7 indicates that the enlarged pose masks yield only marginal improvements in ControlNet, suggesting that the substantial enhancements observed in Stable-Pose are primarily due to the innovative design of PMSA, rather than the additional pose masks input.

Table 7: Ablation study on the effectiveness of PMSA and its ViT design. We show the performance of ControlNet with additional pose masks input, and PMSA with ResNet or ViT as backbone.

| Method | AP ↑ | CAP ↑ | PCE ↓ | FID ↓ | KID ↓ | CLIP-score ↑ |
|---|---|---|---|---|---|---|
| ControlNet | 39.52 | 69.19 | 1.54 | **11.01** | **2.23** | 32.65 |
| ControlNet-PM1 | 39.24 | 68.45 | 1.50 | 11.52 | 2.26 | **32.70** |
| ControlNet-PM2 | 40.73 | 69.27 | **1.49** | 11.63 | 2.24 | 32.67 |
| PMSA w/ ResNet | 45.24 | 70.09 | 1.56 | 13.48 | 2.60 | 32.58 |
| PMSA w/ ViT (Ours) | **48.88** | **70.83** | 1.50 | 11.12 | 2.35 | 32.60 |

Further, to validate the effectiveness of the ViT backbone in PMSA (PMSA w/ ViT), we replaced it with a conventional pose-masked self-attention module operating between residual blocks (PMSA w/ ResNet). We integrated the same pose masks in both configurations to ensure a fair comparison. Table 7 demonstrates that the ViT design in PMSA significantly outperforms the conventional approach. This substantiates the superior capability of ViT to capture long-range, patch-wise interactions among various anatomical parts of human poses to enhance the pose alignment.

**Pose Encoding in Stable-Pose.** To further validate the design of pose encoding in Stable-Pose, we implemented an ablation study by removing either the pose encoder $\beta$ or PMSA-ViT, retaining only one type of pose encoding. The results in Table 8 show that using only PMSA-ViT yields an AP of 36.48, which is expected due to the absence of color-coding information for distinguishing body parts. While using $\beta$ alone increases the AP to 45.03. However, the most significant improvement is observed when integrating both local and global information encoding into the Stable-Pose architecture, achieving the highest AP of 48.88.

Table 8: Ablation study on the pose encoding design in Stable-Pose.

| Method | AP ↑ | CAP ↑ | PCE ↓ | FID ↓ | KID ↓ | CLIP-score ↑ |
|---|---|---|---|---|---|---|
| w/o $\beta$ Encoder | 36.48 | 68.91 | 1.55 | 11.17 | 2.76 | 31.90 |
| w/o PMSA-ViT Encoder | 45.03 | 70.38 | 1.52 | 13.67 | 2.49 | 32.53 |
| w/ both $\beta$ & PMSA-ViT Encoder | 48.88 | 70.83 | 1.50 | 11.12 | 2.35 | 32.60 |

**Pose Masks During Inference**. We incorporate the pose masks during inference by default to enhance pose control. To further validate their effectiveness, we additionally conducted experiments with removing the pose masks during inference. As shown in Table 9, this led to approximately

a 3% drop in AP. This could be due to two main reasons: 1) The pose masks provided additional guidance, thus enhancing control; 2) The inclusion of pose masks maintains consistency between the model's behavior during training and inference. Thus, including pose masks benefits pose control in the generation.

Table 9: Ablation study on the presence of pose masks during inference.

| Method | AP ↑ | CAP ↑ | PCE ↓ | FID ↓ | KID ↓ | CLIP-score ↑ |
|---|---|---|---|---|---|---|
| w/o mask in inference | 45.93 | 70.51 | 1.52 | 13.11 | 2.55 | 32.68 |
| w/ mask in inference | 48.88 | 70.83 | 1.50 | 11.12 | 2.35 | 32.60 |

**Pose-mask Guidance Strength in Loss**. In our proposed loss function in Eq. (7), the parameter $\alpha$, referred to as the pose-mask guidance strength, controls the intensity of penalization applied to the pose regions. We evaluated the impact of varying $\alpha$ from 1 to 10 on pose alignment and image quality, with the results presented in Figure 5. Increasing $\alpha$ in our proposed loss means putting more attention on the foreground pose regions. However, when $\alpha$ is getting too large, it forces the model to learn irrelevant texture information like clothing, which negatively impacts training and slightly decreases AP. Despite this, Stable-Pose still outperforms others across an $\alpha$ range of 1-10. Notably, increasing

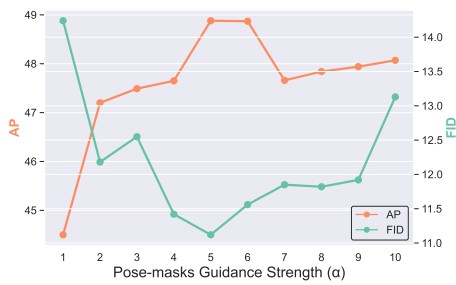

Figure 5: Ablation on pose-mask guidance strength in the loss function.

$\alpha$ has a significant impact on FID, worsening it from 11.0 at $\alpha$=5 to 13.0 at $\alpha$=10. This indicates that focusing solely on the pose regions may decrease the quality of generated content in non-pose regions. Thus there exists a slight trade-off in selecting $\alpha$ to maintain both high pose accuracy and image quality, in which a value around 5 to 6 turns out to be optimal.

## 5 Discussion and Conclusion

We introduced Stable-Pose, a novel adapter that leverages vision transformers with a coarse-to-fine pose-masked self-attention strategy, specifically designed to efficiently manage precise pose controls during T2I generation. Stable-Pose outperforms current controllable T2I generation methods across five distinct public datasets, demonstrating high generation robustness across diverse environments and various pose scenarios. Notably, in complex scenarios involving rare poses such as side or back views and multiple figures, Stable-Pose exhibits exceptional performance in both pose and visual fidelity. This can be attributed to its advanced capability in capturing long-range patch-wise relationships between different anatomical parts of human pose images through our intricate conditioning design. As a result, Stable-Pose holds significant potential in applications demanding high pose accuracy. It can be easily integrated into any pre-trained T2I diffusion models as a lightweight generic adapter to effectively enhance pose control. One limitation of Stable-Pose is its slightly longer inference time (Sec. A.2), primarily due to the integration of self-attention mechanisms within the ViT. In addition, despite excelling in pose control, Stable-Pose has yet to be evaluated with other conditions such as edge maps. Nevertheless, its design allows for straightforward adaptation to various external conditions, suggesting high potential for diverse applications.

**Broader Impacts**: Stable-Pose's excellent pose control makes it a valuable tool in creating diverse artworks, animations, movies, and sports training programs. Additionally, it can be a reliable tool in healthcare and rehabilitation for correcting posture and preventing patients from musculoskeletal issues.

## Acknowledgements

This work was supported by the Munich Center for Machine Learning (MCML) and the German Research Foundation (DFG). The authors gratefully acknowledge the computational and data resources provided by the Leibniz Supercomputing Centre.

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

# A  Appendix / supplementary material

| Index | Keypoint |
|-------|----------|
| 0 | Nose |
| 1 | Left-eye |
| 2 | Right-eye |
| 3 | Left-ear |
| 4 | Right-ear |
| 5 | Left-shoulder |
| 6 | Right-shoulder |
| 7 | Left-elbow |
| 8 | Right-elbow |
| 9 | Left-wrist |
| 10 | Right-wrist |
| 11 | Left-hip |
| 12 | Right-hip |
| 13 | Left-knee |
| 14 | Right-knee |
| 15 | Left-ankle |
| 16 | Right-ankle |

(a) Table of keypoints.  (b) Sample pose with keypoints.  (c) Sample image with pose.

Figure A.1: Illustration of pose input, where each body part is marked in different color. The pose-image pair is from UBC Fashion dataset [43].

## A.1  Datasets and Preprocessing

Our method and the other techniques are trained and evaluated on five distinct datasets as below:

*Human-Art* [15]: The Human-Art dataset comprises 38,000 images distributed across 19 scenarios, encompassing natural scenes, 2D artificial scenarios, and 3D artificial scenarios. We adopt the same train-validation split as the authors suggested. The annotations in Human-Art belong to the International Digital Economy Academy (IDEA) and are licensed under the Attribution-Non Commercial-Share Alike 4.0 International License (CC-BY-NC-SA 4.0).

*LAION-Human* [16]: The LAION-Human is derived from the LAION-5B dataset [36], consisting of approximately 1 million images filtered by human estimation confidence scores. We randomly selected a subset of 200,000 images for training and 20,000 images for validation. The dataset is licensed under the Creative Common CC-BY 4.0 license, which poses no particular restriction. The images are under their copyright.

*UBC Fashion* [43]: The UBC Fashion dataset comprises sequences showcasing fashion models executing turns. We extracted frames representing various orientations: front, side, and back, to rigorously test our model's aptitude for handling both complex and infrequently encountered poses. The dataset yields approximately 1100 front, 450 side, and 1100 back frames. The dataset is licensed under the Creative Commons Attribution-Non Commercial 4.0 International Public License.

*Dance Track* [40]: The Dance Track dataset presents group dance footage, typified by multiple subjects and intricate postures. We curated 20 videos from the Dance Track validation set and extracted a total of 2000 frames to assess our model. The annotations of DanceTrack are licensed under a Creative Commons Attribution 4.0 License and the dataset of DanceTrack is available for non-commercial research purposes only.

*DAVIS* [27]: The DAVIS dataset is a widely used dataset for video-related tasks. We randomly chose 26 human-centric scenarios from the DAVIS Test-Dev 2017 set and the DAVIS Test-Challenge 2017 set, which provided around 2000 frames for evaluation. The DAVIS dataset is released under the BSD License.

All datasets adhere to a standardized protocol featuring 17 keypoints and a maximum of 10 persons per image, following COCO and Human-Art. Despite the original checkpoints of ControlNet and

Table A.1: Trainable parameters (millions), training time on Human-Art (hours), and inference time (seconds per image) for each technique.

| Method | T2I-Adapter | ControlNet | Uni-ControlNet | GLIGEN | HumanSD | Stable-Pose (Ours) |
|---|---|---|---|---|---|---|
| Train. params. | 77 | 361 | 411 | 209 | 859 | 364 |
| Train. time | 12.9 | 14.0 | 14.2 | 20.2 | 19.6 | 14.3 |
| Infer. time | 4.45 | 5.76 | 6.20 | 9.48 | 4.20 | 6.37 |

T2I-Adapter being anchored in the OpenPose keypoint protocol, it is feasible to convert keypoints between different styles without loss of accuracy. In preprocessing the datasets for network input, we applied a score threshold of 0.05 to filter out keypoints and connected corresponding keypoints, following the procedure outlined by the authors of Human-Art. Each connecting line is depicted in distinct colors, as illustrated in Figure A.1, enabling the network to learn associations between each line and specific body parts. Please note that the colors of the pose skeleton presented in this paper are solely for visualization purposes and do not correspond to those used in the experiments. During training, we employed consistent data augmentations, including random cropping at 512 pixels, random rotation, random color shift, and random brightness contrast. These augmentations were applied uniformly across all techniques.

It is worth noting that the scale of the LAION-Human subset and its data distribution closely align with those reported in the SOTA works, such as ControlNet [44], which was trained on 200,000 images sourced from the Internet. Thus, ensuring a fair comparison is possible. Since none of the aforementioned video datasets are annotated with poses or textual descriptions, we employed the GPT-4 API for generating prompts and HigherHRNet [6] for deducing pose labels.

## A.2   Training and Evaluation

To ensure a fair comparison, we trained and evaluated all the techniques on Human-Art dataset. On LAION-Human subset, due to computational constraints, we only trained our method, HumanSD, GLIGEN and Uni-ControlNet, while ControlNet and T2I-Adapter were evaluated using the released checkpoints. The rationale behind this was that HumanSD had been previously trained on a segment of LAION-Human, which might overlap with our validation set, leading to potential data leakage. GLIGEN had been trained with considerably more GPU hours compared to other techniques, posing a risk of an unfair comparison. Uni-ControlNet's available checkpoints are tailored for a multi-condition model, however, our research is dedicated exclusively to pose control, necessitating further training. Conversely, for ControlNet and T2I-Adapter, the amount of data and GPU hours for training were on par with our methods, allowing for fair comparison using the checkpoints they provided.

Training was conducted on two NVIDIA A100 GPUs. Our method required 15 hours to complete training on the Human-Art dataset and 70 hours for the LAION-Human subset. During training, the peak RAM usage for our technique was 25 GB. We reported the number of trainable parameters and training time on Human-Art dataset for each technique in Table A.1. Our technique has comparable number of trainable parameters and training time to ControlNet but produces significantly better results, highlighting the effectiveness of our model design. We also reported the inference speed of each technique in Table A.1, measured by the average seconds needed to generate an image. To ensure a fair comparison, these measurements were conducted on the same NVIDIA A100 GPUs. Due to the self-attention mechanism in the ViT, our method's inference speed is slightly slower than that of ControlNet.

## A.3   Selecting Gaussian Filters in PMSA

A common way to choose the kernel size $k$ of a Gaussian Filter with sigma $\sigma$ is $k = 2 \times \lceil 3\sigma \rceil + 1$. This ensures that the kernel captures the majority of the Gaussian function's weight. When using the kernels and sigmas as specified in the above equation, we observed that the difference between the generated coarse and fine masks was too large to yield satisfactory results. We measured this via the computation of the activation rate, reported in Table A.2. We applied the pose mask to the patches and then calculated the ratio of pose-masked patches to the total number of patches. We opted for

Table A.2: Ratios of unmasked patches (%) under different sigma $\sigma$ and kernel size $k$ in a coarse-to-fine manner.

| $\sigma = 4, k = 25$ | $\sigma = 3, k = 19$ | $\sigma = 2, k = 13$ | $\sigma = 1, k = 7$ |
|:---:|:---:|:---:|:---:|
| 15.83 | 14.12 | 12.38 | 10.48 |

Table A.3: Ratios of unmasked patches (%) under fixed sigma $\sigma = 3$ and varying kernel size $k$ in a coarse-to-fine manner. Selected ones are in bold.

| **23** | 21 | 19 | 17 | 15 | **13** | 11 | **9** | 7 | 5 | 3 | 1 |
|:---:|:---:|:---:|:---:|:---:|:---:|:---:|:---:|:---:|:---:|:---:|:---:|
| 14.26 | 14.17 | 14.12 | 13.98 | 13.77 | 13.21 | 12.78 | 12.14 | 11.53 | 10.81 | 10.06 | 9.09 |

an alternative method to control the blurring effect by fixing $\sigma$ and adjusting $k$, which changes the number of pixels to which the Gaussian weights are applied. Larger values for $k$ dilate the mask, resulting in coarser masks.

In our experiment, we fix $\sigma$ at 3 and then change $k$ for obtaining different kernels. The activation rate of patches under the fixed $\sigma$ and varying $k$ is reported in Table A.3, where $k = 23$ is the largest kernel size for $\sigma = 3$, indicating that any larger $k$ does not affect the activation rate. This size was chosen to generate the most coarse mask. To ensure smooth guidance of our pose-mask in the proposed PMSA, we selected kernel sizes with approximately one percent difference in activation rate. As a result, $k = 13$ was chosen to generate the fine mask. For our ablation study with 3 pose masks, we selected $k = 9$, which also yielded about a one percent difference in activation rate compared to $k = 13$. Details of this ablation study can be found in Sec. A.6.

## A.4  Evaluation Metrics

We define the evaluation metrics in detail here. FID assumes that the features extracted from real and generated images by the Inception v3 model [41] follow a Gaussian distribution. It measures the Fréchet distance between these distributions. KID, however, relaxes the assumption of a Gaussian distribution and calculates the squared Maximum Mean Discrepancy (MMD) between the Inception features of real and generated images using a polynomial kernel. Mean Average Precision (AP) computes the alignment between keypoints in real images and generated images. Poses of generated images are detected by the same Considering that images may contain multiple persons, we included People Counting Error (PCE)[7] as a metric, measuring the false positive rate when generating images featuring multiple people. The CLIP score measures the similarity between embeddings of generated images and text prompts, both of which are encoded by CLIP.

**Challenges in the Current Metrics**. We followed the common metrics widely adopted in the current generative AI field, however, despite the rapid advancements in generative AI, existing metrics have not evolved to provide a more accurate evaluation [14]. Some of the issues are 1) CLIP score relies on cosine similarity between the model's semantic understanding and the given text, which may not align with pose assessments or the relevance of generated images. Additionally, this score is sensitive to the arrangement and composition of elements in images; even minor changes can result in significant fluctuations in the score, which may not accurately reflect the overall generative quality. [34] suggests a Dino-based score; 2) FID estimates the distance between a distribution of Inception-v3 features of real images and those of images generated by the generative models. However, Inception's poor representation of the rich and varied content generated by modern text-to-image models incorrect normality assumptions and poor sample complexity [14]. Thus, the FID score does not account for semantic correctness or content relevance—specifically pose—in relation to the specified text or conditions. Relying solely on FID and CLIP scores does not provide a comprehensive assessment of the generative model. Therefore, we further evaluated our method with a new state-of-the-art metric CMMD [14], which is based on richer CLIP embeddings and the maximum mean discrepancy distance with the Gaussian RBF kernel. It is an unbiased estimator that does not make any assumptions on the probability distribution of the embeddings, offering a more robust and reliable assessment of image quality. As shown in Table A.4, our method achieves better CMMD value compared to HumanSD, demonstrating comparably high image quality.

Table A.4: CMMD evaluation for HumanSD and Stable-Pose.

|  | HumanSD | Stable-Pose |
|---|---|---|
| CMMD ↓ | 5.027 | 5.025 |

Table A.5: Detailed results on UBC Fashion dataset with front, side, and back orientations.

| Orientation | Method | Pose Accuracy | | | Image Quality | | T2I Alignment |
|---|---|---|---|---|---|---|---|
| | | AP ↑ | CAP ↑ | PCE ↓ | FID ↓ | KID ↓ | CLIP-score ↑ |
| Front | T2I-Adapter | 72.20 | 50.16 | 0.61 | 6.57 | 3.95 | 32.11 |
| | ControlNet | 74.64 | 48.11 | 0.65 | **4.50** | **3.91** | 32.06 |
| | Uni-ControlNet | 79.47 | 51.21 | 0.57 | 6.14 | 4.40 | **32.14** |
| | GLIGEN | 73.97 | 50.95 | **0.44** | 8.94 | 4.49 | 30.35 |
| | HumanSD | 76.83 | 51.45 | 0.60 | 6.55 | 4.89 | 31.14 |
| | Stable-Pose (Ours) | **87.26** | **51.90** | 0.56 | 6.05 | 4.57 | 32.05 |
| Side | T2I-Adapter | 36.80 | 44.38 | 0.75 | 6.10 | **3.81** | 32.16 |
| | ControlNet | 52.83 | 46.16 | 0.80 | **5.05** | 3.95 | 32.13 |
| | Uni-ControlNet | 58.26 | 46.70 | 0.70 | 5.73 | 4.28 | 32.15 |
| | GLIGEN | 45.32 | 45.68 | **0.69** | 6.58 | 3.94 | 30.75 |
| | HumanSD | 57.09 | 45.85 | 0.78 | 5.14 | 4.75 | 31.95 |
| | Stable-Pose (Ours) | **69.76** | **47.00** | 0.71 | 5.10 | 4.43 | **32.25** |
| Back | T2I-Adapter | 6.03 | 22.38 | 0.91 | 8.49 | 4.22 | **32.20** |
| | ControlNet | 23.68 | 24.61 | 0.97 | 7.53 | **4.01** | 32.15 |
| | Uni-ControlNet | 19.97 | 23.79 | 0.93 | 7.20 | 4.37 | 32.19 |
| | GLIGEN | 4.45 | 18.36 | **0.90** | 10.83 | 4.32 | 30.29 |
| | HumanSD | 11.05 | 21.15 | 0.91 | 9.12 | 4.99 | 32.01 |
| | Stable-Pose (Ours) | **29.08** | **25.21** | 0.93 | **6.24** | 4.56 | 32.11 |

Table A.6: Detailed results on DAVIS dataset and Dance Track dataset.

| Dataset | Method | Pose Accuracy | | | Image Quality | | T2I Alignment |
|---|---|---|---|---|---|---|---|
| | | AP ↑ | CAP ↑ | PCE ↓ | FID ↓ | KID ↓ | CLIP-score ↑ |
| DAVIS | T2I-Adapter | 20.28 | 60.76 | 1.90 | 3.69 | 2.19 | **31.05** |
| | ControlNet | 30.13 | 60.81 | 1.67 | 3.81 | 2.21 | 30.72 |
| | Uni-ControlNet | 37.64 | 60.57 | 1.56 | 3.60 | 1.83 | 30.63 |
| | GLIGEN | 12.17 | 59.87 | 1.83 | 3.72 | **1.69** | 30.16 |
| | HumanSD | 38.32 | 60.59 | **1.37** | **2.94** | 1.82 | 28.49 |
| | Stable-Pose (Ours) | **42.87** | **62.43** | 1.45 | 3.54 | 1.84 | 30.56 |
| Dance Track | T2I-Adapter | 10.36 | 72.38 | 4.82 | 19.76 | 5.64 | **33.01** |
| | ControlNet | 16.45 | 73.16 | 5.78 | 20.92 | 5.13 | 32.76 |
| | Uni-ControlNet | 25.22 | 73.62 | 4.79 | 21.89 | 4.75 | 32.83 |
| | GLIGEN | 5.61 | 72.57 | 4.72 | 20.13 | **4.53** | 32.37 |
| | HumanSD | 24.13 | 69.93 | **3.78** | 23.15 | 8.58 | 31.64 |
| | Stable-Pose (Ours) | **28.58** | **74.97** | 4.81 | **17.28** | 4.74 | 32.90 |

## A.5 Detailed Results

We reported detailed results for the UBC Fashion, DAVIS, and Dance Track datasets in Tables A.5&A.6, where KID is multiplied by 100 for readability. The results demonstrate that our Stable-Pose consistently provides better controllability over poses, even for challenging poses such as back poses.

## A.6 Additional Ablation Studies

**ViT Parameters**. We studied the impact of different ViT parameters, including depth and patch sizes, as reported in Table A.7 and A.8. Table A.7 shows results for varying depths with a fixed patch size

Table A.7: Comparison of different depths in our proposed ViT. Kernel sizes of Gaussian Filters are noted.

| Depth | AP ↑ | CAP ↑ | PCE ↓ | FID ↓ | KID ↓ | CLIP-score ↑ |
|---|---|---|---|---|---|---|
| 1 (23) | 47.96 | 70.75 | 1.57 | 11.94 | 2.45 | 32.55 |
| 2 (23, 13) | **48.88** | 70.83 | **1.50** | **11.12** | **2.35** | **32.60** |
| 3 (23, 13, 9) | 47.64 | **71.13** | 1.55 | 12.22 | 2.65 | 32.58 |

Table A.8: Comparison of different patch sizes in our proposed ViT.

| Patch size | AP ↑ | CAP ↑ | PCE ↓ | FID ↓ | KID ↓ | CLIP-score ↑ |
|---|---|---|---|---|---|---|
| 2 | **48.88** | 70.83 | **1.50** | **11.12** | **2.35** | 32.60 |
| 4 | 47.51 | **70.85** | 1.53 | 13.39 | 2.49 | 32.58 |
| 8 | 45.69 | 70.44 | 1.59 | 13.14 | 2.47 | **32.64** |

of 2, using coarse-to-fine pose masks as guidance. While these masks provide smooth guidance, a higher depth with overly fine masks might lose valuable information from previous attention layers. Table A.8 presents results for a ViT with PMSA of depth 2 and different patch sizes. Our default patch size of 2 yields the best results, likely because the latent encoding has typically lower dimension (here $64 \times 64$) compared to the dimensions of input high-resolution images. Larger patches may dilute learning of interconnections among anatomical parts for encompassing too much information.

## A.7 Attention Maps of PMSA

In Figure A.2, we present examples of attention maps extracted from our ViT. These maps are derived from the last block of the ViT and represent an average across all attention heads. The attention maps were approximately overlapped with the pose region, which corroborates the objectives of our proposed PMSA. This alignment underscores the efficacy of our model in focusing on relevant pose features, a critical aspect of our approach to improving model interpretability.

## A.8 Failure Case

Stable-Pose may face failure cases when generating the wrong number of people in very crowded scenes. Stable-Pose enhances SD's accuracy in pose-mask regions, whereas a pre-trained SD may still produce human-like figures in the background. For example, in the first row of Figure A.3, Stable-Pose generates an extra half-shaped person on the very right side.

## A.9 Extreme cases

The generation of some extreme pose cases is shown in Figure A.4, such as bending the upper body backward in some dancing poses. As shown in the figures, Stable-Pose still maintains very high pose accuracy on generated images under these challenging scenarios, whereas ControlNet fails to depict the correct pose and body structures.

## A.10 More Visualization Results

We include additional qualitative results from the five datasets we evaluated in Figure A.5,A.6,A.7,A.8, & A.9.

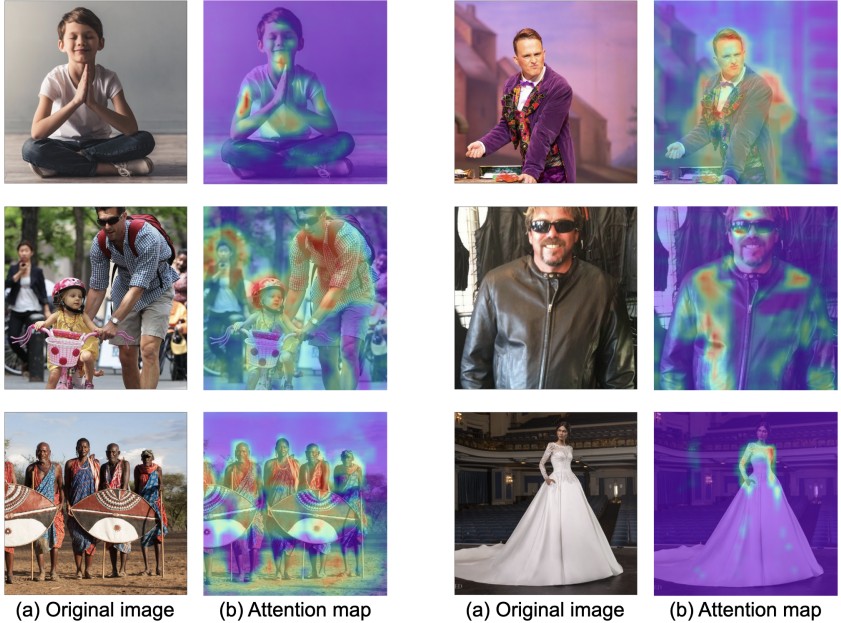

(a) Original image  (b) Attention map    (a) Original image  (b) Attention map

Figure A.2: Samples from LAION-Human dataset with original images and attention maps.

*Ballet performers in white tutus gracefully dancing on stage with a winter background.*

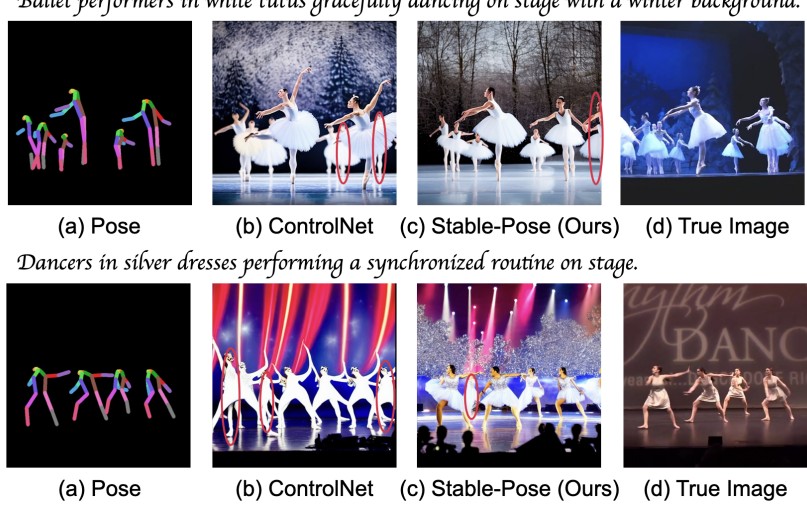

(a) Pose    (b) ControlNet  (c) Stable-Pose (Ours)  (d) True Image

*Dancers in silver dresses performing a synchronized routine on stage.*

(a) Pose    (b) ControlNet  (c) Stable-Pose (Ours)  (d) True Image

Figure A.3: Failure cases sampled from Dance Track dataset. Unexpected humans generated are marked in red circles.

*Acrobatics, a man and a woman performing aerial acrobatics.*

(a) Pose  (b) ControlNet  (c) Stable-Pose (Ours)  (d) True Image

*Acrobatics, a man doing a handstand on a white background.*

(a) Pose  (b) ControlNet  (c) Stable-Pose (Ours)  (d) True Image

Figure A.4: Extreme poses sampled from the Human-Art dataset. ControlNet creates wrong limbs in extreme cases while Stable-Pose achieves accurate target poses.

*Garage kits, a figurine with blue hair and wings.*

*Kids drawing, a painting of a ballerina in blue and yellow.*

*Digital art, rainy day by Kazuhiro Kawai.*

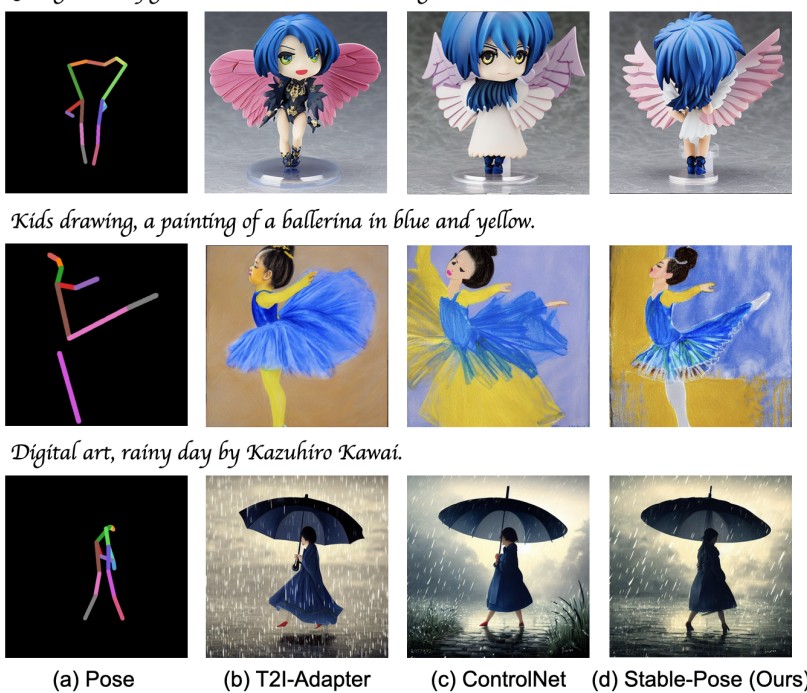

(a) Pose  (b) T2I-Adapter  (c) ControlNet  (d) Stable-Pose (Ours)

Figure A.5: Results on Human-Art dataset.

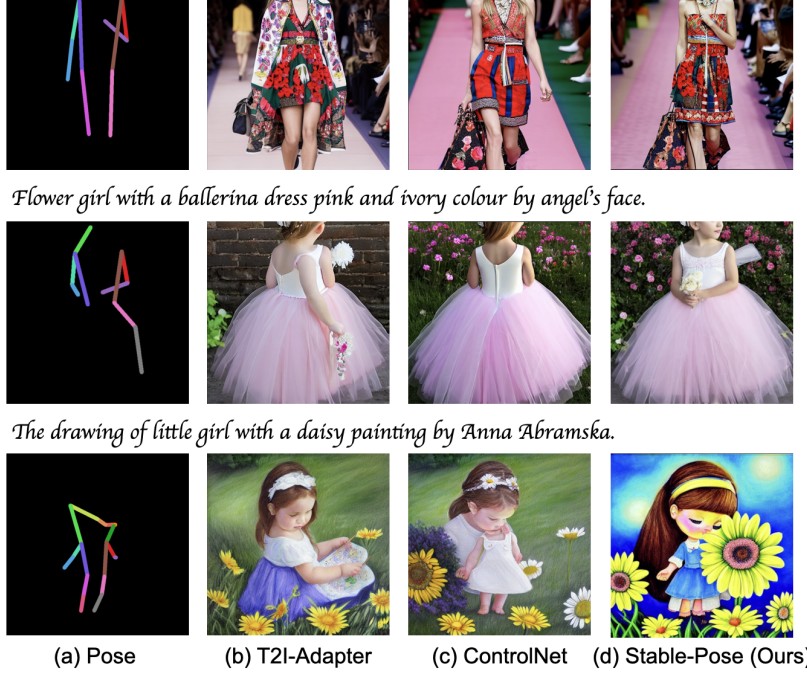

Figure A.6: Results on LAION-Human dataset.

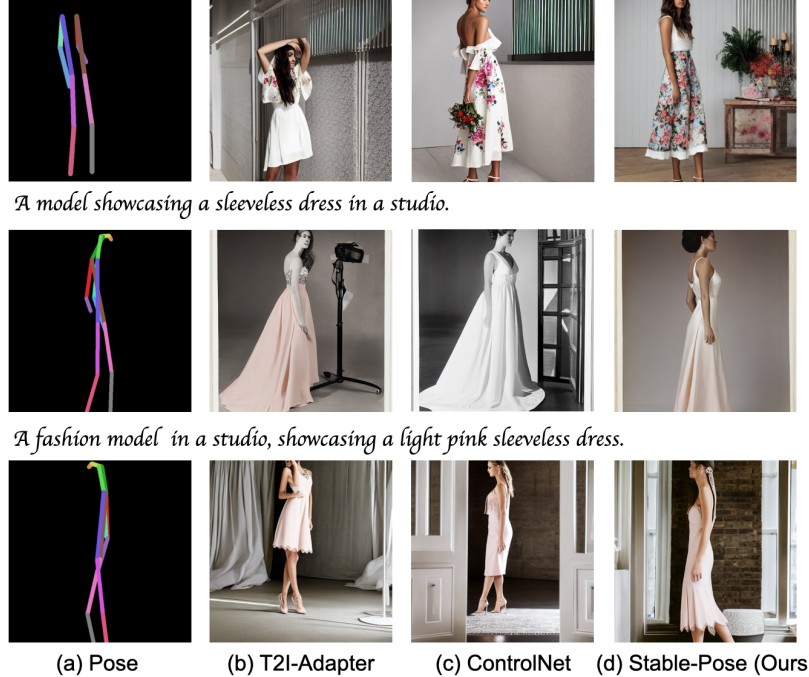

Figure A.7: Results on UBC Fashion dataset with side orientation.

*A model showcasing a floral dress against a plain background.*

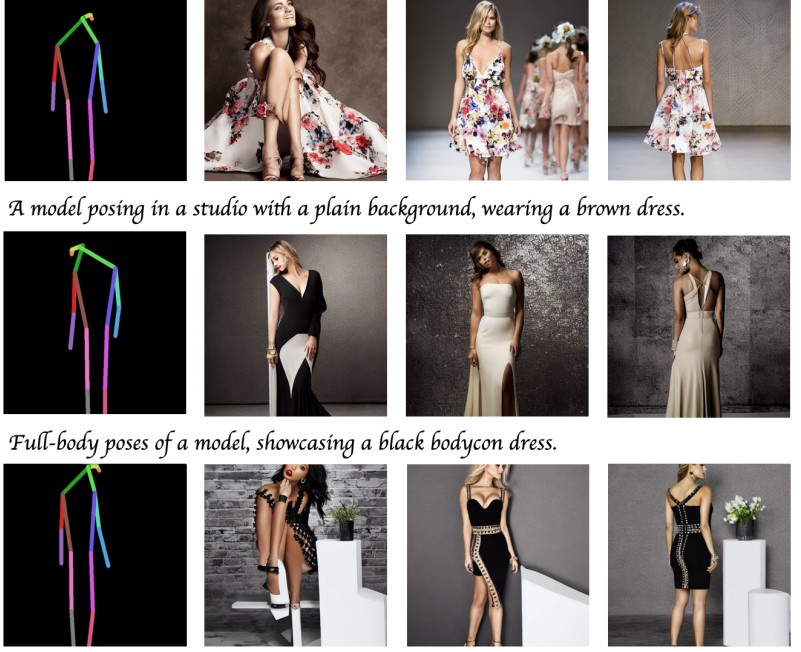

*A model posing in a studio with a plain background, wearing a brown dress.*

*Full-body poses of a model, showcasing a black bodycon dress.*

(a) Pose      (b) T2I-Adapter      (c) ControlNet      (d) Stable-Pose (Ours)

Figure A.8: Results on UBC Fashion dataset with back orientation.

*Dance crew performing synchronized routine with artistic masks.*

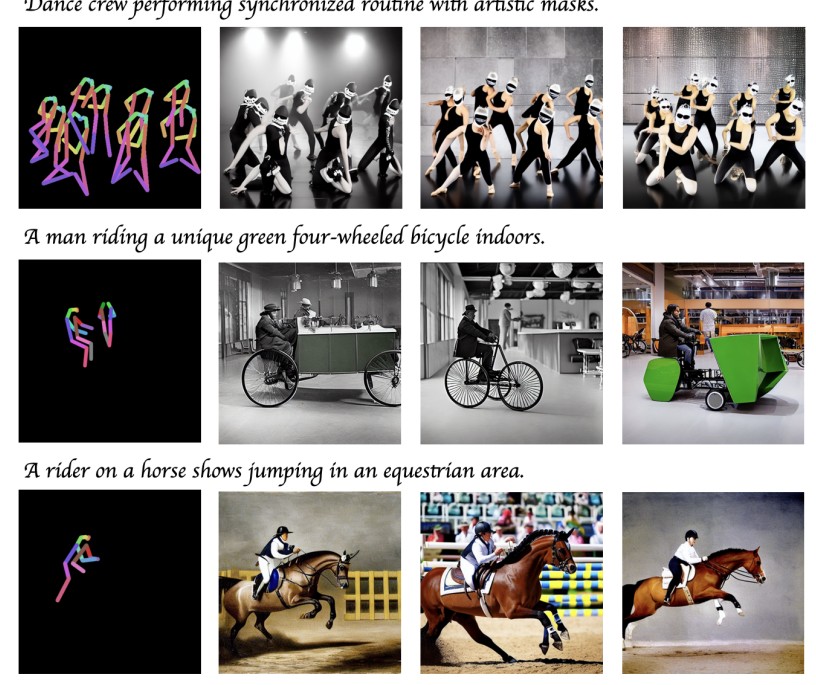

*A man riding a unique green four-wheeled bicycle indoors.*

*A rider on a horse shows jumping in an equestrian area.*

(a) Pose      (b) T2I-Adapter      (c) ControlNet      (d) Stable-Pose (Ours)

Figure A.9: Results on Dance Track (first row) and DAVIS dataset (the rest rows).

