# OpenReview forum: "Stable-Pose: Leveraging Transformers for Pose-Guided Text-to-Image Generation"
_NeurIPS.cc/2024/Conference — NeurIPS 2024 poster_

### Official Review · Reviewer_RKmi · 2024-07-09

**Soundness:** 3
**Presentation:** 3
**Contribution:** 2
**Rating:** 4
**Confidence:** 5

**Summary:**

The paper introduces Stable-Pose, a novel adapter model designed to enhance controllable T2I diffusion models by improving pose guidance, particularly in complex human pose conditions. Utilizing a coarse-to-fine attention masking strategy within a Vision Transformer , Stable-Pose effectively refines pose representation and improves performance over existing methods on the LAION-Human dataset.

**Strengths:**

The paper proposes an adapter model using a coarse-to-fine attention masking strategy within a Vision Transformer (ViT) to significantly enhance pose guidance for text-to-image (T2I) diffusion models. This approach not only refines pose representation but also demonstrates performance improvement over existing methods on the LAION-Human dataset. Additionally, the comprehensive evaluation across multiple datasets underscore the robustness, generalizability, and collaborative potential of the proposed method.

**Weaknesses:**

1. although this method shows good results in condition-following ability, but it has performance drop on image quality and text alignment.
2. compare to controlnet and other pose-guided t2i models, there is no significant contribution both in training recipe and architecture.

**Questions:**

1. It is interesting to see when pose-mask guidance strength is larger than 7, the AP for pose actually got dropped.
2. for all the examples shown in the paper, are they using the same seed for each of the method? (seed is important)

**Limitations:**

Stable-Pose has slightly longer inference times due to its use of self-attention mechanisms within the Vision Transformer (ViT). It has also not been thoroughly evaluated under various conditions, such as edge maps, leaving some robustness aspects unexplored. Further testing is needed to confirm its reliability and effectiveness across diverse scenarios.

---

> ### Author Rebuttal · Authors · 2024-08-06
>
> We thank the reviewer for the insightful comments and feedback. We have addressed each comment carefully and provided a point-by-point response below.
>
> *__C#1__: although this method shows good results…it has performance drop on image quality and text alignment.*
>
> We thank the reviewer for this insightful comment. We followed the common metrics widely adopted in the current generative AI field, however, despite the rapid advancements in generative AI, existing metrics have not evolved to provide a more accurate evaluation [ref1]. We summarize some of the issues below.
> 1) CLIP score: it relies on cosine similarity between the model's semantic understanding and the given text, which may not align with pose assessments or the relevance of generated images. Additionally, this score is sensitive to the arrangement and composition of elements in images; even minor changes can result in significant fluctuations in the score, which may not accurately reflect the overall generative quality. [ref2] suggests a Dino-based score.
> 2) FID: it estimates the distance between a distribution of Inception-v3 features of real images and those of images generated by the generative models. However, Inception's poor representation of the rich and varied content generated by modern text-to-image models incorrect normality assumptions and poor sample complexity [ref1]. Thus, the FID score does not account for semantic correctness or content relevance—specifically pose—in relation to the specified text or conditions.
>
> Relying solely on FID and CLIP scores does not comprehensively assess the generative model. Therefore, in light of your comment and similar concerns from the other Reviewers, we further evaluated our method with a new state-of-the-art metric CMMD [ref1], which is based on richer CLIP embeddings and the maximum mean discrepancy distance with the Gaussian RBF kernel. It is an unbiased estimator that does not make any assumptions on the probability distribution of the embeddings, offering a more robust and reliable assessment of image quality. As shown in the table below, our method achieves better CMMD value compared to HumanSD, demonstrating comparably high image quality.
> ||HumanSD|Stable-Pose|
> |-|-|-|
> |**CMMD↓**|5.027|5.025|
>
> Thanks to this comment, we add a subsection in Appendix A.4 listing the current challenges associated with pose evaluation via these metrics and additional results using CMMD in our final manuscript.
>
> [ref1] Jayasumana et al. Rethinking fid: Towards a better evaluation metric for image generation. In CVPR 2024.
>
> [ref2] Ruiz, et al. Dreambooth: Fine tuning text-to-image diffusion models for subject-driven generation. In CVPR 2023.
>
> ---
>
> *__C#2__: compared to controlnet and other…no significant contribution both in training recipe and architecture.*
>
> We respectfully hold a different perspective on the assessment of our contributions. Stable-Pose is a light-weight **adapter** that can be easily integrated into any pre-trained T2I diffusion models to effectively enhance pose control, even in challenging conditions. It achieves so by synergistically capturing both local pose details with a coarse-to-fine PMSA ViT, and global semantics via a pose encoder. Its novelty and efficiency are confirmed by other reviewers (pWbK), and Reviewer RKmi noted ‘the robustness, generalizability, and collaborative potential of the proposed method’.
>
> To further validate its generalizability, we conducted additional experiments by applying Stable-Pose on top of a pre-trained HumanSD model. As shown in the table below, the inclusion of Stable-Pose considerably improved the baseline HumanSD by over 10% in AP and 12% in KID, highlighting its high generalizability and effectiveness in enhancing both pose control and image quality.
> |Method|AP ↑|CAP ↑|PCE ↓|FID ↓|KID ↓|CLIP score ↑|
> |-|-|-|-|-|-|-|
> |HumanSD|44.57|69.68|1.37|10.03|2.70|32.24|
> |Stable-Pose|48.88|70.83|1.50|11.12|2.35|32.60|
> |HumanSD+Stable-Pose|49.24|71.01|1.42|10.42|2.37|32.16|
>
> ---
>
> *__Q#1__: It is interesting to see when pose-mask guidance strength is larger than 7, the AP for pose actually got dropped.*
>
> We share the reviewer’s perspective on the slight decrease in AP for higher values of the pose-mask guidance strength (α), which is less than 1%. Increasing α in our proposed loss means putting more attention on the foreground pose regions. However, if α is too large, it forces the model to learn irrelevant texture information like clothing, which negatively impacts training and slightly decreases AP. Despite this, Stable-Pose still outperforms others across an α range of 1-10. Notably, increasing α has a significant impact on FID, worsening it from 11.0 at α=5 to 13.0 at α=10. This indicates that focusing solely on the pose regions may decrease the quality of generated content in non-pose regions. Thus there exists a slight trade-off in selecting α to maintain both high pose accuracy and image quality, in which a value around 5 to 6 turns out to be optimal. Thanks for this insightful comment, we will include it in our final manuscript.
>
> ---
>
> *__Q#2__: for all the examples shown in the paper, are they using the same seed for each of the methods? (seed is important)*
>
> Thank you for pointing out this very important aspect in current generative AI. Yes, we indeed always kept the same seed list for all techniques, including ours, during both training and inference time, to ensure fair comparison and reproducibility. We appreciate your attention to this matter and will add this information to our final manuscript to make it more clear. We will also make our code available on GitHub upon acceptance to support the reproducibility of our results.
>
> ---
>
> Once more, we appreciate the Reviewer's insightful suggestions and hope our detailed responses have addressed all concerns. If any further questions arise, we invite the Reviewer to discuss them during the August 7-13 discussion period. We value your comments and look forward to further engagement.

---

> > ### Comment · Reviewer_RKmi · 2024-08-12
> > **Reply**
> >
> > Thanks for the author rebuttal, it address most of my concern. I would like to adjust my rating to borderline but slightly tend to reject side. The method and motivation in this paper are reasonable, but indeed it is kind of incremental compared to the previous method. I believe this paper is good enough to be accepted by a conference in computer vision, but whether it meets the standards for NeurIPS depends on the judgment of the area chair.

---

> > > ### Author Response · Authors · 2024-08-13
> > >
> > > We thank you for the reply; however, we respectfully hold different views. Our work fits the scope of NeurIPS, as it invites submissions that present new and original research on topics such as applications in vision, language, and deep learning (https://nips.cc/Conferences/2024/CallForPapers). We introduced a novel and general **adapter** that can be seamlessly integrated into various pre-trained text-to-image diffusion models like Stable Diffusion, HumanSD, and others to effectively enhance pose control, which is unique and not presented in prior works. While Uni-ControlNet may appear to be an incremental improvement over ControlNet, it introduces a significant new feature: achieving all conditions within a single adapter. Notably, Uni-ControlNet was accepted at NeurIPS 2023, highlighting the value of such innovations in NeurIPS.

---

### Official Review · Reviewer_pWbK · 2024-07-12

**Soundness:** 3
**Presentation:** 4
**Contribution:** 3
**Rating:** 8
**Confidence:** 4

**Summary:**

The authors introduce a novel approach for pose-guided human image generation by developing a new pose encoder block. This block incorporates a coarse-to-fine attention masking strategy within a Vision Transformer architecture, leveraging the self-attention mechanism of Vision Transformers to analyze the interconnections among various anatomical parts of human pose skeletons. Furthermore, their loss function is specifically designed to place greater emphasis on the pose region. The model's performance is assessed both qualitatively and quantitatively, demonstrating promising results.

**Strengths:**

The paper is well-written and has a clear motivation.

It offers an interesting method for pose-guided human image generation. The innovative aspect of the paper is how it encodes pose information into the latent vector during the denoising process, using the PMSA ViT block. The design of this block is novel and seems effective, as demonstrated by the ablation study in Table 4 and the attention maps shown in Figure A.2.

The qualitative and quantitative results look promising, and the ablation study appears thorough.

Overall, this paper is solid and could be considered for acceptance.

**Weaknesses:**

The coarse-to-fine approach does not appear promising based on the ablation study. The improvement from constant variation is negligible.

Limitations and failure cases are not adequately discussed.

**Questions:**

The claim in lines 34-35, stating that pose-guided image generation requires paired human data during training, is not entirely accurate. Previous works, such as [1], [2], [3] etc., do not require paired information and can be trained in unpaired and unsupervised settings, though they do need appearance information. This requirement is not necessarily a disadvantage, as it suits applications like animation from a single image. It would be beneficial to revise this section to reflect these nuances and include the relevant citations.

How do the results appear for extreme poses, such as bending the upper body backward?

[1] A variational u-net for conditional appearance and shape generation. CVPR 2018
[2] Learning realistic human reposing using cyclic self-supervision with 3d shape, pose, and appearance consistency. ICCV 2021
[3]  Towards purely unsupervised disentanglement of appearance and shape for person images generation. 2020

**Limitations:**

Limitations are not adequately discussed.

---

> ### Author Rebuttal · Authors · 2024-08-06
>
> We highly appreciate the Reviewer for the valuable comments and suggestions. We have carefully addressed each comment, providing a point-by-point response below.
>
> *__C#1__: The coarse-to-fine approach does not appear promising…The improvement from constant variation is negligible.*
>
> Thanks for this comment. As the coarse-to-fine approach is one part of the PMSA design, the strength of PMSA-ViT lies in its synergistic integration of the masking algorithm and kernel designs. As shown in Table 4 of our manuscript, using coarse-to-fine pose masks in PMSA improves the baseline results by approximately 7% in AP. Table 5 also highlights the significance of the order of kernels and how it can significantly impact the results. We believe that this coarse-to-fine approach can also be effectively generalized and incorporated into the loss definition of future diffusion models. Thank you again for the feedback and we will include the discussion in this regard in our final manuscript.
>
> ---
>
> *__C#2__: Limitations and failure cases are not adequately discussed.*
>
> Thank you for the comment. We have attached a PDF file that includes some failure cases of our method, shown in Fig. R1, such as the incorrect number of generated humans in very crowded scenes. Given that Stable-Pose is designed to enhance SD's ability to generate accurate content in pose-mask regions, a pre-trained SD may still produce human-like figures in the background. For example, in the first row of Fig. R1 of our attached PDF, Stable-Pose generates an additional half-shaped person in the background on the right side. We will also include a dedicated subsection (A.9) in the Appendix to address this issue with the visualization.
>
> Additionally, in Sec. 5, we discussed some limitations of Stable-Pose, including its slightly longer inference time due to self-attention (illustrated in the appendix Table A.1), and the need for evaluation under other conditions, such as edge maps. Due to the page limit we could not elaborate it extensively in the main manuscript, however, we plan to include deeper discussion on the limitations and failure cases in the final manuscript as there is an additional page available for the improvement of the final submission.
>
> ---
>
> *__Q#1__: The claim…stating that pose-guided image generation requires paired human data during training, is not entirely accurate. Previous works, such as [1], [2], [3] etc., do not require paired information… It would be beneficial to revise this section to reflect these nuances and include the relevant citations.*
>
> Thank you for providing the insightful information and feedback. We agree with the Reviewer that the requirement of paired information is not necessarily a disadvantage in some applications. In our work, we would like to differentiate from this line of methods by focusing on a pose-guided T2I model that offers higher flexibility in training and content generation without constraints on the appearance. We also acknowledge the importance of discussing unsupervised training methods with appearance constraints, as highlighted by the Reviewer. We will include the relevant citations and revise our final manuscript accordingly. Thank you again for your valuable input.
>
> ---
>
> *__Q#2__: How do the results appear for extreme poses, such as bending the upper body backward?*
>
> Thank you for this insightful suggestion. We have illustrated some extreme pose cases in Fig. R2 in our attached PDF file, such as bending the upper body backward in some dancing poses. As shown in the figures, Stable-Pose still maintains very high pose accuracy on generated images under these challenging scenarios, whereas ControlNet fails to depict the correct pose and body structures. We will also add a dedicated subsection (A.10) in the Appendix to address this concern.
>
> ---
>
> We deeply appreciate every comment and suggestion of the Reviewer. We hope we have satisfactorily addressed every point. Your input is valued, and we invite the reviewer to raise any additional questions during the upcoming discussion period from August 7-13.

---

> > ### Comment · Reviewer_pWbK · 2024-08-12
> > **Reply to the authors**
> >
> > Thank you for taking the time to reply to my queries. I have carefully read the concerns about the other authors and the rebuttal. I still think this is a good paper to get accepted and would support it.

---

> > > ### Author Response · Authors · 2024-08-12
> > >
> > > Thank you for your thoughtful feedback and support. We appreciate your careful consideration of the concerns and are glad to hear you view the paper positively.

---

### Official Review · Reviewer_qFuv · 2024-07-13

**Soundness:** 3
**Presentation:** 3
**Contribution:** 2
**Rating:** 5
**Confidence:** 4

**Summary:**

The authors proposed a pose guided image generation pipeline. By introducing the PMSA ViT as pose encoder, the pipeline can generate more controllable images with corresponding 2D pose conditions.

**Strengths:**

- According to Fig 4, the proposed method is the only one which distinguish the front and back side of the 2D pose condition, which is much better than other SOTAs.
- The generated images follows the pose condition better than other methods.

**Weaknesses:**

- The proposed method is more like an incremental work of previous methods instead of a new innovative pipeline.
- Although the generated images follow the pose guidance better, the image quality(FID) and T2I Alignment(CLIP Score) are worse than those SOTA methods.

**Questions:**

- Could the authors discuss more about why image quality(FID) and T2I Alignment(CLIP Score) are worse than those SOTA methods?
- Why the proposed methods are better at distinguish the left and right of the pose condition? During PSMA ViT, it seems like the authors treated all the keypoints the same, without any special design to improve the left-right consistency.

**Limitations:**

The limitation has been discussed in the paper.

---

> ### Author Rebuttal · Authors · 2024-08-06
>
> We appreciate the Reviewer’s comments and insightful suggestions. Below, we provide a point-by-point response to each comment.
>
> *__C#1__: The proposed method is more like an incremental work…instead of a new innovative pipeline.*
>
> We respectfully hold a different perspective on the novelty assessment. Stable-Pose is a light-weight **adapter** that can be easily integrated into any pre-trained T2I diffusion models to effectively enhance pose control, even in challenging conditions. It achieves so by synergistically capturing both local pose details with a coarse-to-fine PMSA ViT, and global semantics via a pose encoder. Its novelty and efficiency are confirmed by other reviewers (pWbK), and Reviewer RKmi noted ‘the robustness, generalizability, and collaborative potential of the proposed method’.
>
> To further validate its generalizability, we conducted additional experiments by applying Stable-Pose on top of a pre-trained HumanSD model. As shown in the table below, the inclusion of Stable-Pose considerably improved the baseline HumanSD by over 10% in AP and 12% in KID, highlighting its high generalizability and effectiveness in enhancing both pose control and image quality.
> |Method|AP ↑|CAP ↑|PCE ↓|FID ↓|KID ↓|CLIP score ↑|
> |-|-|-|-|-|-|-|
> |HumanSD|44.57|69.68|1.37|10.03|2.70|32.24|
> |Stable-Pose|48.88|70.83|1.50|11.12|2.35|32.60|
> |HumanSD+Stable-Pose|49.24|71.01|1.42|10.42|2.37|32.16|
>
> ---
>
> *__C#2__: Although the generated images follow the pose guidance better, the image quality (FID) and T2I Alignment (CLIP Score) are worse…*
>
> *& __Q#1__: Could the authors discuss more about why image quality (FID) and T2I Alignment (CLIP Score) are worse than those SOTA methods?*
>
> We thank the reviewer for this insightful comment. We followed the common metrics widely adopted in the current generative AI field, however, despite the rapid advancements in generative AI, existing metrics have not evolved to provide a more accurate evaluation [ref1]. We summarize some of the issues below.
> 1) CLIP score: it relies on cosine similarity between the model's semantic understanding and the given text, which may not align with pose assessments or the relevance of generated images. Additionally, this score is sensitive to the arrangement and composition of elements in images; even minor changes can result in significant fluctuations in the score, which may not accurately reflect the overall generative quality. [ref2] suggests a Dino-based score.
> 2) FID: it estimates the distance between a distribution of Inception-v3 features of real images and those of images generated by the generative models. However, Inception's poor representation of the rich and varied content generated by modern text-to-image models incorrect normality assumptions and poor sample complexity [ref1]. Thus, the FID score does not account for semantic correctness or content relevance—specifically pose—in relation to the specified text or conditions.
>
> Relying solely on FID and CLIP scores does not provide a comprehensive assessment of the generative model. Therefore, in light of your comment and similar concerns from the other Reviewers, we further evaluated our method with a new state-of-the-art metric CMMD [ref1], which is based on richer CLIP embeddings and the maximum mean discrepancy distance with the Gaussian RBF kernel. It is an unbiased estimator that does not make any assumptions on the probability distribution of the embeddings, offering a more robust and reliable assessment of image quality. As shown in the table below, our method achieves better CMMD value compared to HumanSD, demonstrating comparably high image quality.
> ||HumanSD|Stable-Pose|
> |--|--|--|
> |**CMMD ↓**|5.027|5.025|
>
> Thanks to this comment, we add a subsection in Appendix A.4 listing the current challenges associated with pose assessment via these metrics and the additional results using CMMD in our final manuscript.
>
> [ref1] Jayasumana et al. "Rethinking fid: Towards a better evaluation metric for image generation." In CVPR 2024.
>
> [ref2] Ruiz, et al. "Dreambooth: Fine tuning text-to-image diffusion models for subject-driven generation. In CVPR 2023.
>
> ---
>
> *__Q#2__: Why the proposed methods are better at distinguish the left and right of the pose condition?*
>
> Thanks for this comment. Indeed, the PMSA-ViT focuses on exploring relationships between human parts using Transformers, capturing local pose details without differentiating body parts. Yet simultaneously, Stable-Pose includes a pose encoder (β) to encode color-coded pose skeleton information, capturing semantic details like limb positions and the number of people. This enables Stable-Pose to naturally and flexibly enhance pose accuracy by synergistically integrating both local pose details and global human conditions.
>
> To further validate the design, we specifically add an ablation study by removing either β or PMSA-ViT, retaining only one type of encoding. The results in the table below show that using only PMSA-ViT yields an AP of 36.48, which is expected due to the absence of color-coding information for distinguishing body parts. While using β alone increases the AP to 45.03. However, the most significant improvement is observed when integrating both local and global information encoding into the Stable-Pose architecture, achieving the highest AP of 48.88.
> |Method|AP ↑|CAP ↑|PCE ↓|FID ↓|KID ↓|CLIP score ↑|
> |-|-|-|-|-|-|-|
> |w/o β Enc.|36.48|68.91|1.55|11.17|2.76|31.90|
> |w/o PMSA-ViT Enc.|45.03|70.38|1.52|13.67|2.49|32.53|
> |w/ both β & PMSA-ViT Encs.|48.88|70.83|1.50|11.12|2.35|32.60|
>
> Thanks to this comment, we add these results and discussion to our final manuscript and we appreciate this valuable discussion.
>
> ---
>
> We thank you once again for every comment. Our responses aim to address all concerns. If there are further questions or suggestions, we invite the Reviewer to discuss them during the August 7-13 reviewer-author period. We value your input and look forward to productive discussions.

---

### Official Review · Reviewer_ypoD · 2024-07-13

**Soundness:** 2
**Presentation:** 3
**Contribution:** 2
**Rating:** 5
**Confidence:** 4

**Summary:**

To obtain more refined pose condition control, especially for challenging condition (pose & text) generation, this paper designs a coarse-to-fine Pose-Masked Self-Attention (PMSA) module with the use of pose masks, fine-tunes Stable Diffusion with pose-mask guided loss, and ultimately achieves more precise control with little sacrifice of visual generation quality.

**Strengths:**

- The paper shows a good accurate control effect consistent with the pose and text guidance.

- The method makes sense because pose is a more sophisticated control condition than text, so it requires a refined and localized design. It is simple and relatively easy to understand.

- There are some experimental verifications in terms of controllability and robustness.

**Weaknesses:**

I carefully evaluated this module paper and did not find particular flaws except for no consideration of the multi-person condition. I will refer to the review suggestions of other reviewers and the author's feedback later.

[Method]

- (No consideration of different persons & parts) Surprisingly, the conditions may include multiple person, but PMSA does not explicitly consider modeling different people and different parts (in this sense the DensePose is a better choice), but only models the relationship between patches and semantics are considered by high-level pose decoder $\beta$. (Although the qualitative results shown look fine, there is no wrong association, such as A's hand is generated at B's hand position, or A's hand is generated at A's foot) I hope the author could provide more detailed explanations and discussion, even on the potential improvement to the proposed PMSA.

[Experiments]

- (Inferior PCE) Can the authors try to analyze the disadvantages of PCE (People Count Error) of the image generated by this method (Tab. 1)?

- (Add module architecture ablation studies) Since this is a work that proposes a module, the ablation studies of the module structure may still be helpful, e.g. the number of the PMSA-ViT block.

- (Add FPS results) Because the proposed PMSA is a plugin, to give readers a better comprehension, it is recommended that the author report the inference time (e.g. FPS) together with other metric results (although the author's claim method has disadvantages).

- (Add failures) It is also important to understand the boundaries of a method, e.g. the failure cases. It is recommended to visualize and analyze some of them.

- (Opinions on masked generation) Pose-mask guided is used as training loss in the paper. To further enhance control, it seems that the generation of humans can be similarly replaced within the mask area during the inference time. I am curious about the author’s opinion on this.

**Questions:**

Please see the weaknesses.

**Limitations:**

The authors state the limitations in Sec. 5 as 1. longer inference time caused by the Self-Attention, 2. not evaluated with other condition guidances. It is recommended to add the visualization and discussion of failure cases.

---

> ### Author Rebuttal · Authors · 2024-08-06
>
> We express our sincere appreciation to the reviewer for providing valuable comments and suggestions. We have addressed the comments point-by-point, as outlined below.
>
> *__C#1__: (No consideration of different persons & parts)*
>
> Thanks for the insightful discussion. The pose information is encoded in two ways in Stable-Pose:
> 1) *The pose encoder (β)*: It encodes the RGB pose skeleton input into high-level contextual information. Through color encoding, β captures semantic information such as the differentiation of limbs and the number of people.
> 2) *PMSA-ViT*: the patchified latent features are processed through several coarse-to-fine pose-masked self-attentive ViTs, which focus on accurately modeling the poses shown in the pose skeleton and the interaction between different pose parts.
>
> Thus, through these two modules, Stable-Pose synergistically integrates both local and global contextual information (including the number of people and body parts) from the pose skeleton naturally.
>
> To further validate the design, we specifically add an ablation study by removing either β or PMSA-ViT, retaining only one type of encoding. The results in the table below show that using only PMSA-ViT yields an AP of 36.48, which is expected due to the absence of color-coding information for distinguishing body parts. While using β alone increases the AP to 45.03. However, the most significant improvement is observed when integrating both local and global information encoding into the Stable-Pose architecture, achieving the highest AP of 48.88.
> |Method|AP ↑|CAP ↑|PCE ↓|FID ↓|KID ↓|CLIP score ↑|
> |-|-|-|-|-|-|-|
> |w/o β Enc.|36.48|68.91|1.55|11.17|2.76|31.90|
> |w/o PMSA-ViT Enc.|45.03|70.38|1.52|13.67|2.49|32.53|
> |w/ both β & PMSA-ViT Encs.|48.88|70.83|1.50|11.12|2.35|32.60|
>
> Thanks to this comment, we will add these results and discussion to our final manuscript and we appreciate this valuable comment.
>
> ---
>
> *__C#2__: (Inferior PCE)*
>
> While PCE is useful for evaluating the number of humans in generated images, it has limitations such as the sensitivity to outliers. Given that Stable-Pose is designed to enhance SD's ability to generate accurate content in pose-mask regions, a pre-trained SD may still produce human-like figures in the background. For example, in the first row of Fig. R1 of our attached PDF, Stable-Pose generates an additional half-shaped person in the background on the right side. Such cases affect the overall PCE measurement. Therefore, foreground counting should be prioritized over background counting. However, PCE treats both equally.
>
> Despite this, Stable-Pose still reaches the best PCE score among adapter-based methods (e.g., ControlNet) in LAION-Human. In addition, applying Stable-Pose on top of a pre-trained HumanSD model (a fine-tuning-based method) considerably improves the PCE score compared to using Stable-Pose with SD. While it results in a marginal error of only 0.05 more/less people per image compared to the HumanSD method, it substantially improves the pose accuracy in AP by 10%.
> |Method|AP ↑|CAP ↑|PCE ↓|FID ↓|KID ↓|CLIP score ↑|
> |-|-|-|-|-|-|-|
> |HumanSD|44.57|69.68|1.37|10.03|2.70|32.24|
> |Stable-Pose|48.88|70.83|1.50|11.12|2.35|32.60|
> |HumanSD+Stable-Pose|49.24|71.01|1.42|10.42|2.37|32.16|
>
> Thanks to this comment, we further discuss this aspect of PCE in Appendix A.4 and would love to hear your opinion about it.
>
> ---
>
> *__C#3__: (Add module architecture ablation studies) & __C#4__: (Add FPS results)*
>
> Thanks for your advice in improving the paper. In fact we have addressed these concerns in our supplementary material as detailed below:
> 1) In Table A.6 and A.7, we demonstrated ablation studies of the module structure, including the number of the PMSA-ViT blocks and different patch sizes in the ViT, respectively.
> 2) In Table A.1, we reported the inference time (FPS) and training duration (in hours) of our method compared to the other state-of-the-art approaches.
>
> Due to the page limit we could not include these results in the main manuscript, but only in the appendix. That said, we plan to move some parts mentioned above to the main draft, as there is an additional page available for the final submission.
>
> ---
>
> *__C#5__: (Add failures)*
>
> Thank you for your insightful comment. We attached a PDF file with some failure cases in Fig. R1, such as generating the wrong number of people in very crowded scenes. Stable-Pose enhances SD's accuracy in pose-mask regions, but a pre-trained SD may still produce human-like figures in the background. For example, in the first row of Fig. R1, Stable-Pose generates an extra half-shaped person on the very right side. We will add a dedicated subsection (A.9) in the Appendix to address this issue.
>
> ---
>
> *__C#6__: (Opinions on masked generation)*
>
> We highly appreciate this insightful comment. In our implementation, we indeed incorporate the pose masks during inference by default to enhance control. To further validate their effectiveness, we additionally conducted experiments with removing the pose masks during inference. As shown in the table below, this led to approximately a 3% drop in AP. This could be due to two main reasons:
> 1. The pose masks provided additional guidance, thus enhancing control.
> 2. The inclusion of pose masks maintains consistency between the model's behavior during training and inference.
>
> |Method|AP ↑|CAP ↑|PCE ↓|FID ↓ |KID ↓|CLIP score ↑|
> |-|-|-|-|-|-|-|
> |w/o mask in inference|45.93|70.51|1.52|13.11|2.55|32.68|
> |w/ mask in inference|48.88|70.83|1.50|11.12|2.35|32.60|
>
> Thus, including pose masks benefits pose control in the generation. Thanks to this comment, we will include this result in our final manuscript.
>
> ---
>
> We highly appreciate the valuable suggestions and input from the Reviewer. We hope our responses address the reviewer’s concerns about Stable-Pose, with details covered in the manuscript and appendix. We look forward to any further questions during the reviewer-author discussion period from August 7-13.

---

> > ### Comment · Reviewer_ypoD · 2024-08-14
> >
> > I carefully read the reviewers' reviews and the author's careful and detailed rebuttal. Incluing the ablation study of local and global semantic modules, the experiments that can be combined with other backbones for plug-and-play, and the experimental supplement of masking inference. I want to further follow up and add some:
> >
> > (**C#1**) The author does not seem to answer me directly. Have you observed failures in associating joints with the wrong person when using implicit conditional encoding color? Or does the author think that the current local and global designs are sufficient to avoid this problem? Will explicit consideration of the encoding of different people bring further gains?
> >
> > (**C#2**) The author is suggested to add PCE or other quantitative evaluation experimental results of the foreground person number in future editions to verify the qualitative background false positive speculation.

---

> > > ### Author Response · Authors · 2024-08-14
> > >
> > > We highly appreciate the efforts of the reviewer and would like to answer the follow-up questions as below:
> > >
> > > **C#1**: We did not observe significant failures in associating joints with the wrong person, particularly when compared to other SOTA methods. We believe that our current local and global designs are sufficient for effectively modeling multiple individuals. Furthermore, we consider that explicitly encoding different people would not significantly enhance the performance in this context. For example, GLIGEN is a method that explicitly learns keypoint embeddings for different body parts across different individuals. However, it did not yield promising results in terms of APs or PCEs. Additionally, the explicit modeling approach in GLIGEN substantially increases the training time compared to other methods.
> > >
> > > **C#2**: Thank you for the suggestion. We agree with adding quantitative results in the foreground area and will work on reducing the influence of false positives generated in the background in our future release.

---

### Author Rebuttal · Authors · 2024-08-06

We sincerely thank the Reviewers for their insightful and constructive feedback. We have incorporated their suggestions and responded comprehensively to their comments. We have addressed the comments of the Reviewers individually. We have incorporated a Q&A section in the Supplementary file that concisely summarizes the discussion related to specific questions raised by the Reviewers. We hope that our study will be well-received as a valuable contribution to the NeurIPS' focus on theory & application. We are available for any further discussions or inquiries the reviewers may have during the reviewer-author discussion period.

It is important to mention that the attached PDF file includes figures that have been prepared to address some of the comments from two respected reviewers with the IDs: ypoD and pWbK.

Best regards,

Authors

---

### Comment · Area_Chair_xYM1 · 2024-08-10
**Reviewer-author discussions**

Dear Reviewers ypoD, qFuv, pWbK and RKmi

Would you please look at the rebuttal, discuss with the authors and finalize your score after discussion?

Thanks,

Your AC

---

> ### Comment · Reviewer_pWbK · 2024-08-12
> **Reply to the AC**
>
> I have carefully reviewed the other reviewers' evaluations and the full rebuttal from the authors. As I mentioned earlier, the innovative aspect of the paper lies in how it encodes pose information into the diffusion model. The results, including those presented in the rebuttal, demonstrate that their proposed method outperforms the state-of-the-art in adhering to that control signal. I agree with the other reviewers that this approach may cause a marginal degradation in performance in terms of FID evaluation. However, I do not believe that having an FID value approximately 1 point higher than other methods is a significant issue that should penalize the proposed method. In fact, I doubt this difference is even perceptible visually, and there could be several factors contributing to it. Overall, I believe the module proposed by the authors provides effective pose control, which requires a certain level of innovation, and there are sufficient experiments to support its efficacy. Therefore, I would support the acceptance of this paper. But of course, I am open to discussion if there are different opinions.

---

### Decision · Program_Chairs · 2024-09-25

**Decision:**

Accept (poster)

**Comment:**

This paper designs a coarse-to-fine pose-masked self-attention strategy and proposes a pose-guided image generation pipeline. After rebuttal, it received SA, 2 BA, and BR.

The initial concerns focused on the design of the conditions (ypoD), performance (qFuv, RKmi), extreme cases (pWbK) and novelty (RKmi).
After going through the paper, the review, and the response, the AC values the key idea of the coarse-to-fine attention masking strategy with ViT, thinks most concerns have been addressed, and recommends the acceptance of the paper. The AC also urges the authors to append the conditions and the performance discussion, and highlight the difference compared to existing works in the revision.